# Indole produced during dysbiosis mediates host–microorganism chemical communication

**Rui-Qiu Yang[1†], Yong-Hong Chen[1†], Qin-yi Wu[2†], Jie Tang[1,3†], Shan-Zhuang Niu[1], Qiu Zhao[1], Yi-Cheng Ma[1]\*, Cheng-Gang Zou[1]\***

[1]State key Laboratory for Conservation and Utilization of Bio-Resources in Yunnan, Yunnan University, Kunming, China; [2]Yunnan Provincial Key Laboratory of Molecular Biology for Sinomedicine, Yunnan University of Traditional Chinese Medicine, Kunming, China; [3]Yunnan Key Laboratory of Vaccine Research Development on Severe Infectious Disease, Chinese Academy of Medical Sciences and Peking Union Medical College, Kunming, China

**Abstract** An imbalance of the gut microbiota, termed dysbiosis, has a substantial impact on host physiology. However, the mechanism by which host deals with gut dysbiosis to maintain fitness remains largely unknown. In *Caenorhabditis elegans*, *Escherichia coli*, which is its bacterial diet, proliferates in its intestinal lumen during aging. Here, we demonstrate that progressive intestinal proliferation of *E. coli* activates the transcription factor DAF-16, which is required for maintenance of longevity and organismal fitness in worms with age. DAF-16 up-regulates two lysozymes *lys-7* and *lys-8*, thus limiting the bacterial accumulation in the gut of worms during aging. During dysbiosis, the levels of indole produced by *E. coli* are increased in worms. Indole is involved in the activation of DAF-16 by TRPA-1 in neurons of worms. Our finding demonstrates that indole functions as a microbial signal of gut dysbiosis to promote fitness of the host.

**\*For correspondence:**
mayc@ynu.edu.cn (Y-ChengM);
chgzou@ynu.edu.cn (C-GangZ)

**\*These authors contributed equally to this work.**

**Competing interest:** The authors declare that no competing interests exist.

## Editor's evaluation

This fundamental study provides compelling evidence for a new mechanism of host-microbe interaction, with indole, produced by proliferating bacteria in the *C. elegans* digestive system, signalling through the host via the transcription factor DAF-16 to induce the expression of genes controlling bacterial growth in the gut. The work is relevant to a wide audience as it invites deeper research into this mechanism, while also serving as a template for similar microbiome/host interactions in other systems.

## Introduction

The microbiota in the gut has a substantial impact on host nutrition, metabolism, immune function, development, behavior, and lifespan (*Bana and Cabreiro, 2019*; *Johnson and Foster, 2018*; *Lee and Brey, 2013*). Microbial community disequilibria, so-called dysbiosis, have been implicated in a broad range of human diseases, such as obesity, insulin resistance, autoimmune disorders, inflammatory bowel disease (IBD), aging, and increased pathogen susceptibility (*Honda and Littman, 2012*; *Wu et al., 2015*). Therefore, understanding of the mechanisms that modulate host–microbe interactions will provide important insights into treatment of these diseases by intervening the microbial communities.

The genetically tractable model organism *Caenorhabditis elegans* has contributed greatly to understand the role of host–microbiota interactions in host physiology (*Cabreiro and Gems, 2013*; *Zhang et al., 2017*). In the bacterivore nematode, the microbiota in the gut can be easily manipulated, making it an excellent model for studying how the microbiota affects host physiology in the context of disease and aging at a single species (*Cabreiro and Gems, 2013*). For instance, the neurotransmitter tyramine produced by intestinal Providencia bacteria can direct sensory behavioral decisions by modulate multiple monoaminergic pathways in *C. elegans* (*O'Donnell et al., 2020*). On the other hand, *C. elegans*-based studies have revealed a variety of signaling cascades involved in the innate immune responses to microbial infection (*Irazoqui et al., 2010b*), including the p38 mitogen-activated protein kinase (MAPK)/PMK-1, ERK MAPK/MPK-1, the heat shock transcription factor (HSF-1), the transforming growth factor (TGF)β/bone morphogenetic protein (BMP) signaling (*Zugasti and Ewbank, 2009*), and the forkhead transcription factor DAF-16/FOXO pathway (*Garsin et al., 2003*; *Kim et al., 2002*; *Singh and Aballay, 2006*; *Zou et al., 2013*). In *Drosophila* and mammalian cells, activation of FOXOs up-regulates a set of antimicrobial peptides, such as drosomycin and defensins (*Becker et al., 2010*), implicating that the role for FOXOs in innate immunity is conserved across species. Furthermore, disruption of these innate immune-related pathways, such as the p38 MAPK pathway and the TGF-β/BMP signaling cascade, turns beneficial bacteria commensal to pathogenic in worms (*Berg et al., 2019*; *Montalvo-Katz et al., 2013*). These results indicate that the integrity of immune system is also essential for host defense against non-pathogenic bacteria.

*Escherichia coli* (strain OP50) is conventionally used as a bacterial food for culturing *C. elegans*. In general, most of *E. coli* is efficiently disrupted by a muscular grinder in the pharynx of the worm. However, very few intact bacteria may escape from this defense system, and enter in the lumen of the worm intestine (*Gupta and Singh, 2017*). The intestinal lumen of worms is frequently distended during aging, which accompanied by bacterial proliferation (*Garigan et al., 2002*; *McGee et al., 2011*). Blockage of bacterial proliferation by treatment of UV, antibiotics, and heat extends lifespan of worms (*De Arras et al., 2014*; *Garigan et al., 2002*; *Hwang et al., 2014*), implicating that progressive intestinal proliferation of *E. coli* probably contributes to worm aging and death. Thus, age-related dysbiosis in *C. elegans* provides a model to study how host responses to altered gut microbiota to maintain fitness (*Ezcurra, 2018*).

Accumulating evidence has indicated that genetic inactivation of *daf-16* accelerates tissue deterioration and shortens lifespan of wild-type (WT) worms grown on *E. coli* OP50 (*Garigan et al., 2002*; *Li et al., 2019*; *Portal-Celhay and Blaser, 2012*; *Portal-Celhay et al., 2012*). The observation that DAF-16 is activated during bacterial accumulation in older worms (*Li et al., 2019*) prompt us to investigate the role of DAF-16 in dysbiosis in the gut of worms. We found that activation of DAF-16 was required for maintenance of longevity and organismal fitness in worms, at least in part, by up-regulating two lysozyme genes (*lys-7* and *lys-8*), thus limiting bacterial accumulation in the gut of worms during aging. Meanwhile, we identified that indole produced by *E. coli* was involved in the activation of DAF-16 by TRPA-1.

## Results

### Activation of DAF-16 is required for normal lifespan and organismal fitness in worms

To study the role of DAF-16 in age-related dysbiosis, all the experiments started from the young adult stage, which was considered day 0 (0 day) (*Figure 1—figure supplement 1A*). Consistent with a recent observation that DAF-16 is activated in worms with age (*Li et al., 2019*), we found that DAF-16::GFP was mainly located in the cytoplasm of the intestine in worms expressing *daf-16p::daf-16::gfp* fed live *E. coli* OP50 on day 1 (*Figure 1A, B*). The nuclear translocation of DAF-16 in the intestine was increased in worms fed live *E. coli* OP50 on days 4 and 7, but not in age-matched WT worms fed heat-killed (HK) *E. coli* OP50 (*Figure 1A, B*). To further confirm these results, we tested the expression of two DAF-16 target genes *dod-3* and *hsp-16.2* via the transcriptional reporter strains of *dod-3p::gfp* and *hsp-16.2p::nCherry*. As expected, the expression of either *dod-3p::gfp* or *hsp-16.2p::nCherry* was significantly up-regulated in worms fed live *E. coli* OP50 on day 4, but not in age-matched worms fed HK *E. coli* OP50 (*Figure 1—figure supplement 1B–D*). Likewise, DAF-16 was also retained in the cytoplasm of the intestine in worms fed ampicillin-killed *E. coli* OP50 on days 4 and 7 (*Figure 1C*). In

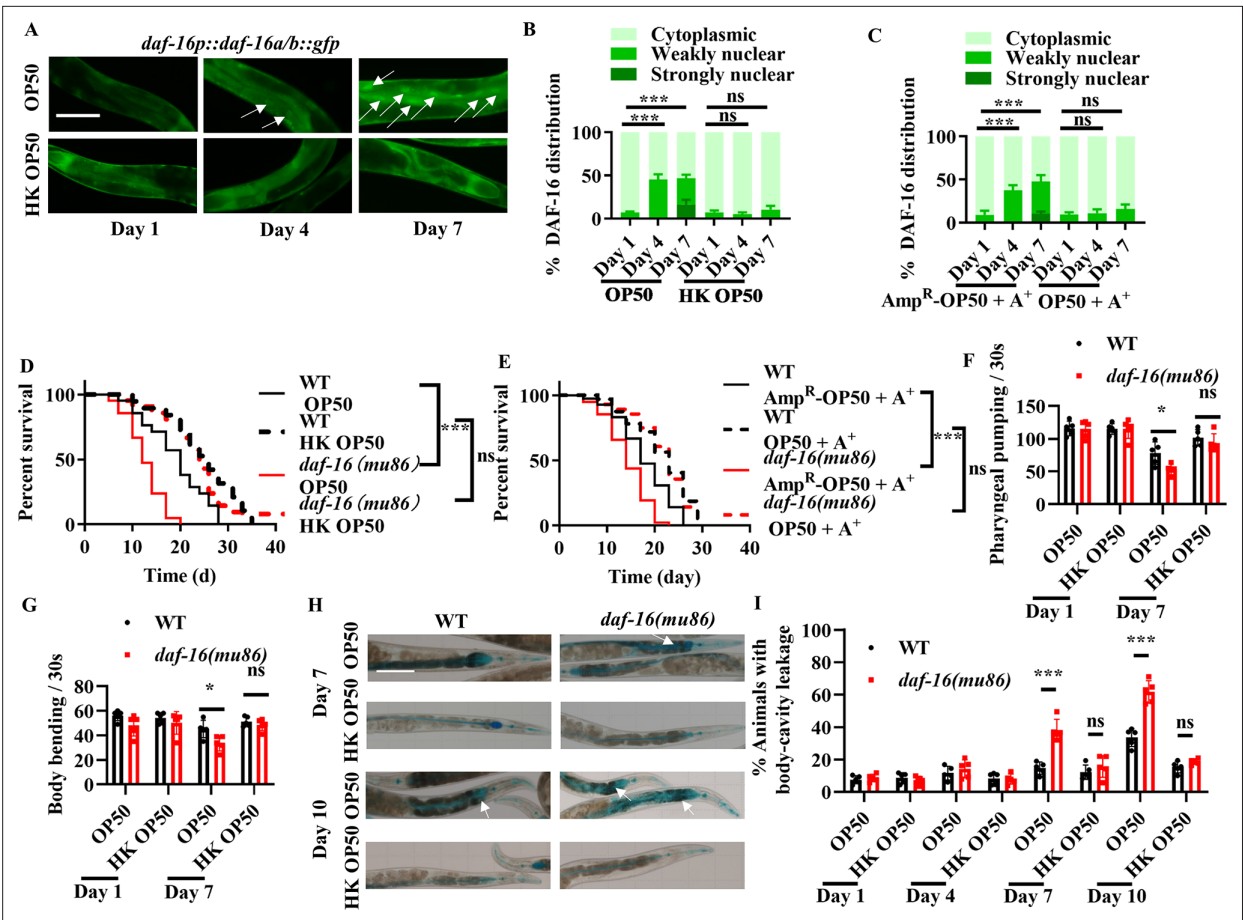

**Figure 1.** Activation of DAF-16 by bacterial accumulation is involved in longevity and organismal fitness. (**A**) The nuclear translocation of DAF-16::GFP in the intestine was increased in worms fed live *E. coli* OP50, but not in those fed heat-killed (HK) *E. coli* OP50. White arrows indicate nuclear localization of DAF-16::GFP. Scale bars: 50 µm. (**B**) Quantification of DAF-16 nuclear localization. These results are means ± standard deviation (SD) of three independent experiments (*n* > 35 worms per experiment). ***p < 0.001. (**C**) DAF-16 was also retained in the cytoplasm of the intestine of worms fed ampicillin-killed *E. coli* OP50 on days 4 and 7. These results are means ± SD of three independent experiments (*n* > 35 worms per experiment). ***p < 0.001. p-values (**B, C**) were calculated using the Chi-square test. (**D**) *daf-16(mu86)* mutants grown on live *E. coli* OP50 had a shorter lifespan than those grown on HK *E. coli* OP50. ***p < 0.001. ns, not significant. (**E**) *daf-16(mu86)* mutants grown on live *E. coli* OP50 had a shorter lifespan than those grown on ampicillin-killed *E. coli* OP50. ***p < 0.001. ns, not significant. A⁺, Ampicillin treatment; Ampᴿ-OP50, *E. coli* OP50 containing an ampicillin resistance plasmid (PMF440). p-values (**D, E**) were calculated using log-rank test. (**F, G**) DAF-16 was involved in delaying the appearance of the aging markers, including pharyngeal pumping (**F**) and body bending (**G**), in worms fed live *E. coli* OP50, but not in those fed HK *E. coli* OP50. These results are means ± SD of five independent experiments (*n* > 20 worms per experiment). *p < 0.05. ns, not significant. (**H**) Representative images of intestinal permeability stained by food dye FD&C Blue No. 1 in worms. White arrows indicate the body-cavity leakages of worms. Scale bars: 100 µm. (**I**) Quantification of body-cavity leakages in animals fed on live *E. coli* OP50 or HK *E. coli* OP50 over time. These results are means ± SD of five independent experiments (*n* > 20 worms per experiment). ***p < 0.001. ns, not significant. p-values (**F, G, I**) were calculated using a two-tailed *t*-test.

The online version of this article includes the following source data and figure supplement(s) for figure 1:

**Source data 1.** Lifespan assays summary and quantification results.

**Figure supplement 1.** The expression of DAF-16 target genes *dod-3* and *hsp-16.2* is significantly up-regulated in worms fed live *E. coli* OP50 on day 4.

**Figure supplement 1—source data 1.** Quantification results.

contrast, starvation induced the nuclear translocation of DAF-16 in the intestine of worms on day 1 (***Figure 1—figure supplement 1E***). Thus, either HK or antibiotic-killed *E. coli* OP50 as a food source does not induce a starvation state in worms. Taken together, these results indicate that the activation of DAF-16 is mainly attributed to the presence of live *E. coli*, but not by age itself, in worms.

Consistent with the previous observations (***Li et al., 2019***; ***Portal-Celhay et al., 2012***), we found that a mutation in *daf-16(mu86)* shortened the lifespan of worms fed live *E. coli* OP50 at 20°C (***Figure 1D***). In contrast, the mutation in *daf-16(mu86)* had no impact on the lifespan of worms fed either HK

(*Figure 1D*) or ampicillin-killed *E. coli* OP50 (*Figure 1E*). Next, we examined the effects of DAF-16 on phenotypic traits, such as the pharyngeal-pumping rate, body bending, and integrity of intestinal barrier, which are associated with aging in worms. Both the rates of pharyngeal pumping (*Figure 1F*) and body bending (*Figure 1G*) were reduced in *daf-16(mu86)* mutants on day 7 as compared to those in WT worms fed live *E. coli* OP50. In contrast, the rates of pharyngeal pumping and body bending were comparable in WT worms and *daf-16(mu86)* mutants grown on HK *E. coli* OP50 on day 7. Furthermore, we used food dye FD&C Blue No. 1 to evaluate the integrity of intestinal barrier (*Ma et al., 2020*). The body-cavity leakage in *daf-16(mu86)* mutants on days 7 and 10 were higher than those in age-matched WT worms fed live *E. coli* OP50, but not HK *E. coli* OP50 (*Figure 1H, I*). Taken together, these results demonstrate that the activation of DAF-16 by bacterial accumulation is required for normal lifespan and the maintenance of organismal fitness.

## Indole produced from *E. coli* activates DAF-16

As DAF-16 is activated by bacterial accumulation, we hypothesized that this activation is probably due to bacterially produced compounds. To test this idea, culture supernatants from *E. coli* OP50 were collected, and isolated by high-performance liquid chromatograph (HPLC), silica gel G column and Sephadex LH-20 column chromatography. A candidate compound was detected by activity-guided isolation, and further identified as indole with mass spectrometry and NMR data (*Figure 2A*, *Figure 2—figure supplement 1A, B*; *Supplementary file 1*). The observation that indole secreted by *E. coli* OP50 could activate DAF-16 was further confirmed by analyzing commercial HPLC grade indole. Supplementation with indole (50–200 µM) not only significantly induced the nuclear translocation of DAF-16 (*Figure 2B*), but also up-regulated the expression of either *dod-3p::gfp* or *hsp-16.2p::nCherry* in young adult worms after 24 hr of treatment (*Figure 2—figure supplement 2A, B*). Next, we found that the levels of indole were 30.9, 71.9, and 105.9 nmol/g dry weight, respectively, in worms fed live *E. coli* OP50 on days 1, 4, and 7 (*Figure 2C*). The elevated indole levels in worms were accompanied by an increase in colony-forming units (CFU) of live *E. coli* OP50 in the intestine of worms with age (*Figure 2C*), suggesting that accumulation of live *E. coli* OP50 in the intestine is probably responsible for increased indole in worms. These data also raised a possibility that exogenous indole produced by *E. coli* OP50 on the nematode growth media (NGM) plates could increase the levels of indole in worms with age. However, we found that the levels of indole were 28.2, 31.6, and 36.1 nmol/g dry weight, respectively, in worms fed HK *E. coli* OP50 on days 1, 4, and 7 (*Figure 2—figure supplement 3A*), indicating that indole was not accumulated in worms fed HK *E. coli* OP50 even for 7 days. Thus, the increase in the levels of indole in worms results from intestinal accumulation of live *E. coli* OP50, rather than exogenous indole produced by *E. coli* OP50 on the NGM plates. The observation that DAF-16 was retained in the cytoplasm of the intestine in worms fed live *E. coli* OP50 on day 1 (*Figure 1A, B*) also indicated that exogenous indole produced by *E. coli* OP50 on the NGM plates is not enough to activate DAF-16. Supplementation with indole (50–200 µM) significantly increased the indole levels in young adult worms on day 1 (*Figure 2—figure supplement 3B*), which could induce nuclear translocation of DAF-16 in worms (*Figure 2B*).

In bacteria, indole is biosynthesized from tryptophan by tryptophanase (*tnaA*) (*Lee et al., 2015*). To determine the effect of indole produced by bacteria in the gut, worms were fed *E. coli* K-12 BW25113 strain (called K-12), and *tnaA*-deficient strain BW25113 *tnaA* (called K-12 *ΔtnaA*), respectively. We found that both *tnaA* mRNA and indole levels were undetectable in the K-12 *ΔtnaA* strain (*Figure 2—figure supplement 4A and B*). Furthermore, disruption of *tnaA* significantly suppressed the nuclear translocation of DAF-16 (*Figure 2D*). The nuclear translocation of DAF-16::GFP was mainly located in the cytoplasm of the intestine in worms fed live K-12 *ΔtnaA* strains on day 4. However, supplementation with indole induced the nuclear translocation of DAF-16::GFP in the intestine of these worms (*Figure 2—figure supplement 4C*). Moreover, we rescued the expression of *tnaA* in the K-12*ΔtnaA* strains, and found that the production of indole in the K12*ΔtnaA::tnaA* strains bacterial solution was 34.1 µmol/l, similar to the K12 strains (*Figure 2—figure supplement 4D*). DAF-16::GFP was mainly located in the nuclear of the intestine in worms fed live K12*ΔtnaA::tnaA* strains on days 4 and 7 (*Figure 2—figure supplement 4E*). Taken together, our results suggest that indole is involved in the activation of DAF-16 in worms with age.

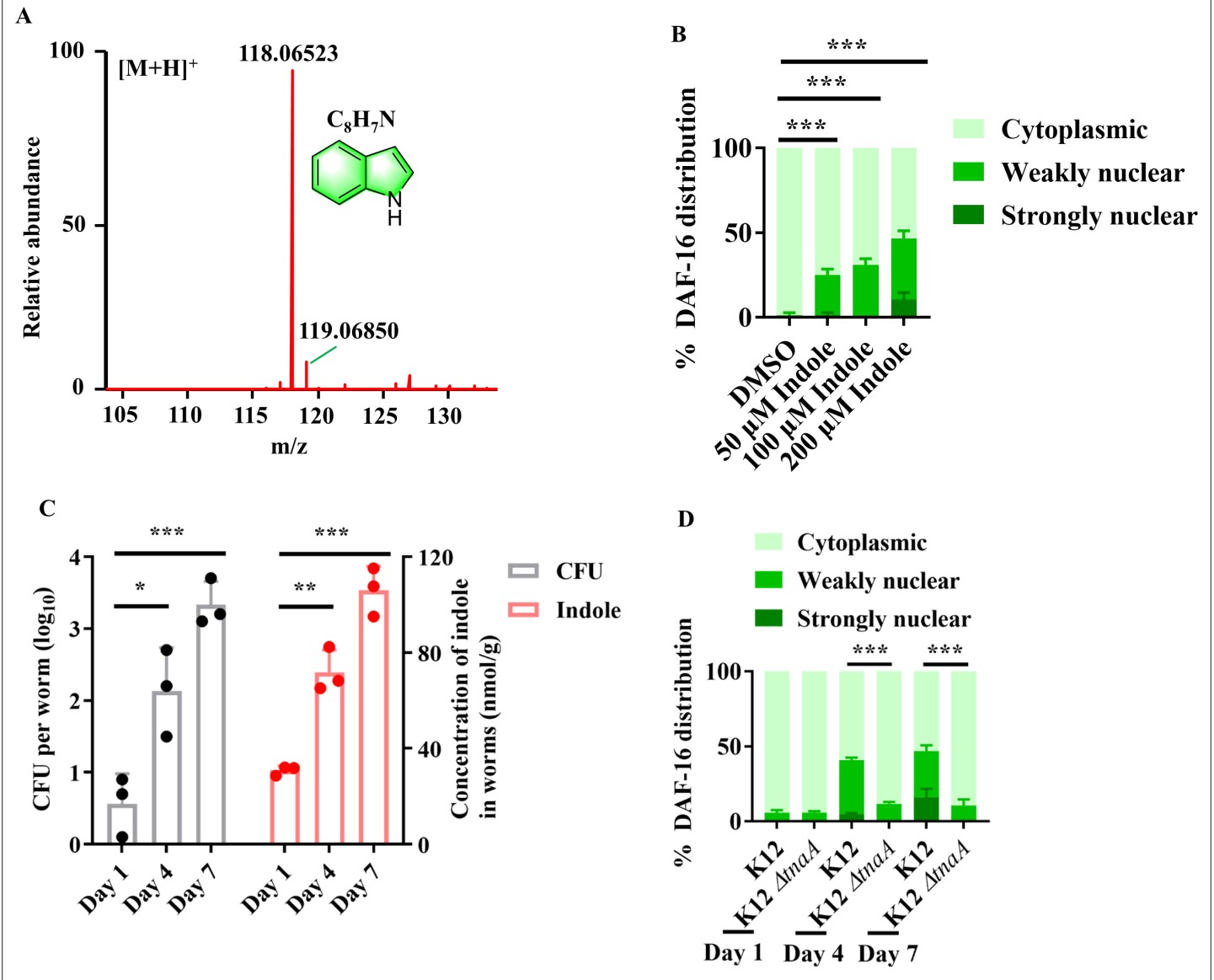

**Figure 2.** Indole is involved in the nuclear translocation of DAF-16 in worms with age. (**A**) High-resolution mass spectrum of indole. (**B**) Supplementation with indole promoted the nuclear translocation of DAF-16::GFP in the intestine of worms. These results are means ± standard deviation (SD) of three independent experiments (*n* > 35 worms per experiment). ***p < 0.001. (**C**) Colony-forming units (CFU) of *E. coli* OP50 were increased in worms over time, which was accompanied by an increase in the levels of indole in worms. These results are means ± SD of three independent experiments (*n* > 30 worms per experiment). *p < 0.05; **p < 0.01; ***p < 0.001. p-values (**C**) were calculated using a two-tailed *t*-test. (**D**) Deletion of *tnaA* significantly suppressed the nuclear translocation of DAF-16::GFP in the intestine of worms fed *E. coli* BW25113. These results are means ± SD of three independent experiments (*n* > 35 worms per experiment). ***p < 0.001. p-values (**B, D**) were calculated using the Chi-square test.

The online version of this article includes the following source data and figure supplement(s) for figure 2:

**Source data 1.** Quantification results.

**Figure supplement 1.** Indole is the active compound for activation of DAF-16.

**Figure supplement 2.** Indole treatment induces the expressions of DAF-16 target genes in worms.

**Figure supplement 2—source data 1.** Quantification results.

**Figure supplement 3.** Quantitative analysis of indole in *C. elegans* by LC–MS.

**Figure supplement 3—source data 1.** Quantification results.

**Figure supplement 4.** Functional validation of *tnaA*-deficient BW25113 strain.

**Figure supplement 4—source data 1.** Quantification results.

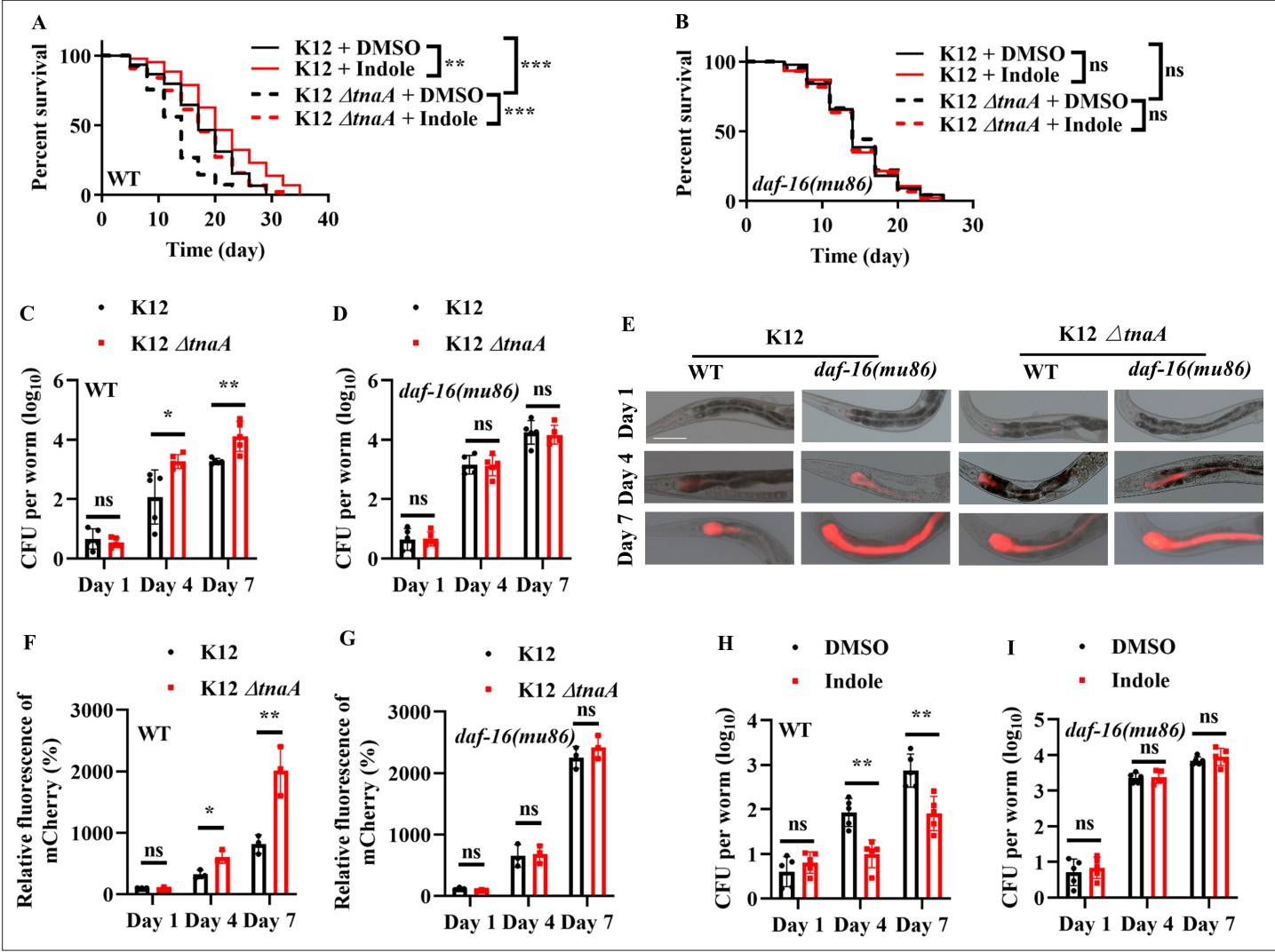

**Figure 3.** Indole is required for maintenance of normal lifespan via DAF-16 in worms. (**A**) Wild-type (WT) worms fed *E. coli* K-12 *tnaA* strains had a shorter lifespan than those fed *E. coli* K-12 strain at 20°C. Supplementation with indole (100 μM) extended the lifespan of WT worms fed *E. coli* K-12, and rescued the short lifespan of WT worms fed *E. coli* K-12 *ΔtnaA* strain. **p < 0.01; ***p < 0.001. (**B**) Indole-mediated lifespan extension depended on DAF-16 in worms. ns, not significant. p-values (**A, B**) were calculated using log-rank test. (**C, D**) Colony-forming units (CFU) of *E. coli* K-12 or K-12*tnaA* were measured in WT worms (**C**) or *daf-16(mu86)* mutants (**D**). These results are means ± standard deviation (SD) of five independent experiments (*n* > 30 worms per experiment). *p < 0.05; **p < 0.01. ns, not significant. (**E**) Fluorescence images of worms exposed to *E. coli* K-12 or K-12*tnaA* expressing mCherry. Scale bars: 50 μm. (**F, G**) Quantification of fluorescent intensity of *E. coli* K-12 or K-12*tnaA* expressing mCherry in WT worms (**F**) or *daf-16(mu86)* mutants (**G**). These results are means ± SD of three independent experiments (*n* > 35 worms per experiment). *p < 0.05; **p < 0.01. ns, not significant. CFU of *E. coli* K-12 were measured in WT worms (**H**) or *daf-16(mu86)* mutants (**I**) in the presence of exogenous indole (100 μM). These results are means ± SD of five independent experiments (*n* > 30 worms per experiment). **p < 0.01. ns, not significant. p-values (**C, D, F–I**) were calculated using a two-tailed *t*-test.

The online version of this article includes the following source data for figure 3:

**Source data 1.** Lifespan assays summary and quantification results.

## Indole produced by bacteria in the gut is required for normal lifespan

It has been shown that exogenous indole extends lifespan in *C. elegans* at 16°C (*Sonowal et al., 2017*). We found that adult worms fed *E. coli* K-12 *tnaA* strains exhibited a shortened lifespan at 20°C, compared with those fed *E. coli* K-12 strain (*Figure 3A*). Supplementation with indole in adults not only rescued the shortened lifespan of WT worms fed *E. coli* K-12 *ΔtnaA* strain, but also significantly extended the lifespan of WT worms fed *E. coli* K-12 strain (*Figure 3A*). In contrast, the lifespan of *daf-16(mu86)* mutants fed *E. coli* K-12 *ΔtnaA* strain was comparable to that of *daf-16(mu86)* mutants

fed *E. coli* K-12 strain (*Figure 3B*). Supplementation with indole (100 μM) did not affect the lifespan of *daf-16(mu86)* mutants fed either *E. coli* K-12 or K-12 *tnaA* strain (*Figure 3B*). Moreover, the CFU of *E. coli* K-12 *ΔnaA* strain were significantly higher than those of *E. coli* K12 strain in WT worms on days 4 and 7 (*Figure 3C*). In contrast, the CFU of *E. coli* K-12 *ΔtnaA* strain were similar to those of *E. coli* K12 strain in *daf-16(mu86)* mutants on days 4 and 7 (*Figure 3D*). Likewise, the accumulation of *E. coli* K-12 *ΔtnaA* strain expressing mCherry was significantly higher than that of *E. coli* K12 strain in WT worms, but not *daf-16(mu86)* mutants, on days 4 and 7 (*Figure 3E–G*). Finally, supplementation with indole (100 μM) inhibited the CFU of *E. coli* K-12 in WT worms, but not *daf-16(mu86)* mutants, on days 4 and 7 (*Figure 3H and I*). These results suggest that indole produced by bacteria in the gut inhibits the proliferation of *E. coli* through DAF-16, thereby maintaining normal lifespan in worms.

## TRPA-1 in neurons is required for indole-mediated longevity

How did worms detect bacterially produced indole during aging? Previously, Sonowal et al. have identified that *C. elegans* xenobiotic receptor AHR-1, which encodes an ortholog of the mammalian AHR, mediates indole-promoted lifespan extension in worms at 16°C (*Sonowal et al., 2017*). However, we found that RNAi knockdown of *ahr-1* did not affect the nuclear translocation of DAF-16 in worms fed *E. coli* K12 strain on day 7 (*Figure 4—figure supplement 1A*) or young adult worms treated with indole (100 μM) for 24 hr (*Figure 4—figure supplement 1B*). A recent study has demonstrated that bacteria-derived indole activates the transient receptor potential ankyrin 1 (TRPA-1), a cold-sensitive TRP channel, in enteroendocrine cells in zebrafish and mammals (*Ye et al., 2021*). As *C. elegans* TRPA-1 is an ortholog of mammalian TRPA-1 (*Kindt et al., 2007*; *Venkatachalam and Montell, 2007*), we tested the role of TRPA-1 in DAF-16 activation by indole. We found that RNAi knockdown of *trpa-1* significantly inhibited the nuclear translocation of DAF-16 in worms fed *E. coli* K12 strain on days 4 and 7 (*Figure 4A*) or young adult worms treated with indole (100 μM) for 24 hr (*Figure 4B*). These results suggest that TRPA-1 is involved in indole-mediated DAF-16 activation. Previously, *Xiao et al., 2013* have demonstrated that TRPA-1 activated by low temperatures mediates calcium influx, which in turn stimulates the PKC-2-SGK-1 signaling to promote the transcription activity of DAF-16, leading to lifespan extension in worms. However, we found that RNAi knockdown of *sgk-1* did not influence the nuclear-cytoplasmic distribution of DAF-16 in the presence of indole (*Figure 4—figure supplement 1C*). Mutant worms lacking *trpa-1* exhibited a shorter lifespan than did WT worms at 20°C (*Xiao et al., 2013*). Consistent with this observation, knockdown of *trpa-1* by RNAi significantly shortened the lifespan of worms fed *E. coli* K12 strain (*Figure 4C*). Supplementation with indole no longer extended the lifespan of worms after knockdown of *trpa-1* by RNAi or in *trpa-1(ok999)* mutants (*Figure 4C*; *Figure 4—figure supplement 2A*). In *trpa-1(ok999)* mutants, knockdown of *daf-16* by RNAi did not further reduce the lifespan of worms (*Figure 4—figure supplement 2A*). Furthermore, the CFU of *E. coli* K-12 strain were significantly increased in worms subjected to *trpa-1* RNAi on days 4 and 7 (*Figure 4D, E*). Supplementation with indole failed to suppress the increases in CFU in these worms. In addition, supplementation with indole also failed to suppress the increases in *E. coli* K-12 strain expressing mCherry in *trpa-1(ok999)* mutants subjected to empty vector or *daf-16* RNAi (*Figure 4—figure supplement 2B*). These results suggest that indole exhibits its function in extending the lifespan and inhibiting bacterial accumulation primarily via TRPA-1. It has been shown that podocarpic acid, a TRPA-1 agonist, activates the SEK-1/PMK-1/SKN-1 pathway (*Chaudhuri et al., 2016*), a signaling cascade involved in *C. elegans* defense against pathogenic bacteria (*Kim et al., 2002*). However, we found that supplementation with 0.1 mM indole failed to induce nuclear localization of SKN-1::GFP in the intestine of the transgenic worms expressing *skn-1p::skn-1::gfp* (*Figure 4—figure supplement 3A, B*), suggesting that indole cannot activate SKN-1 in worms. Furthermore, 0.02 mM of podocarpic acid did not induce the nuclear localization of DAF-16::GFP (*Figure 4—figure supplement 3C*). These results suggest that TRPA-1 activation via podocarpic acid and indole happens largely through distinct mechanisms.

As overexpression of *trpa-1* in both the intestine and neurons is sufficient to extend the lifespan of worms (*Xiao et al., 2013*), we determined tissue-specific activities of TRPA-1 in the regulation of longevity mediated by indole. Consistent with this observation (*Xiao et al., 2013*), we found that both neuronal- and intestinal-specific knockdown of *trpa-1* by RNAi significantly shortened the lifespan of worms (*Figure 5A, B*). However, supplementation with indole (100 μM) only extended the lifespan of worms subjected to intestinal specific (*Figure 5B*), but not neuronal specific, *trpa-1* RNAi (*Figure 5A*).

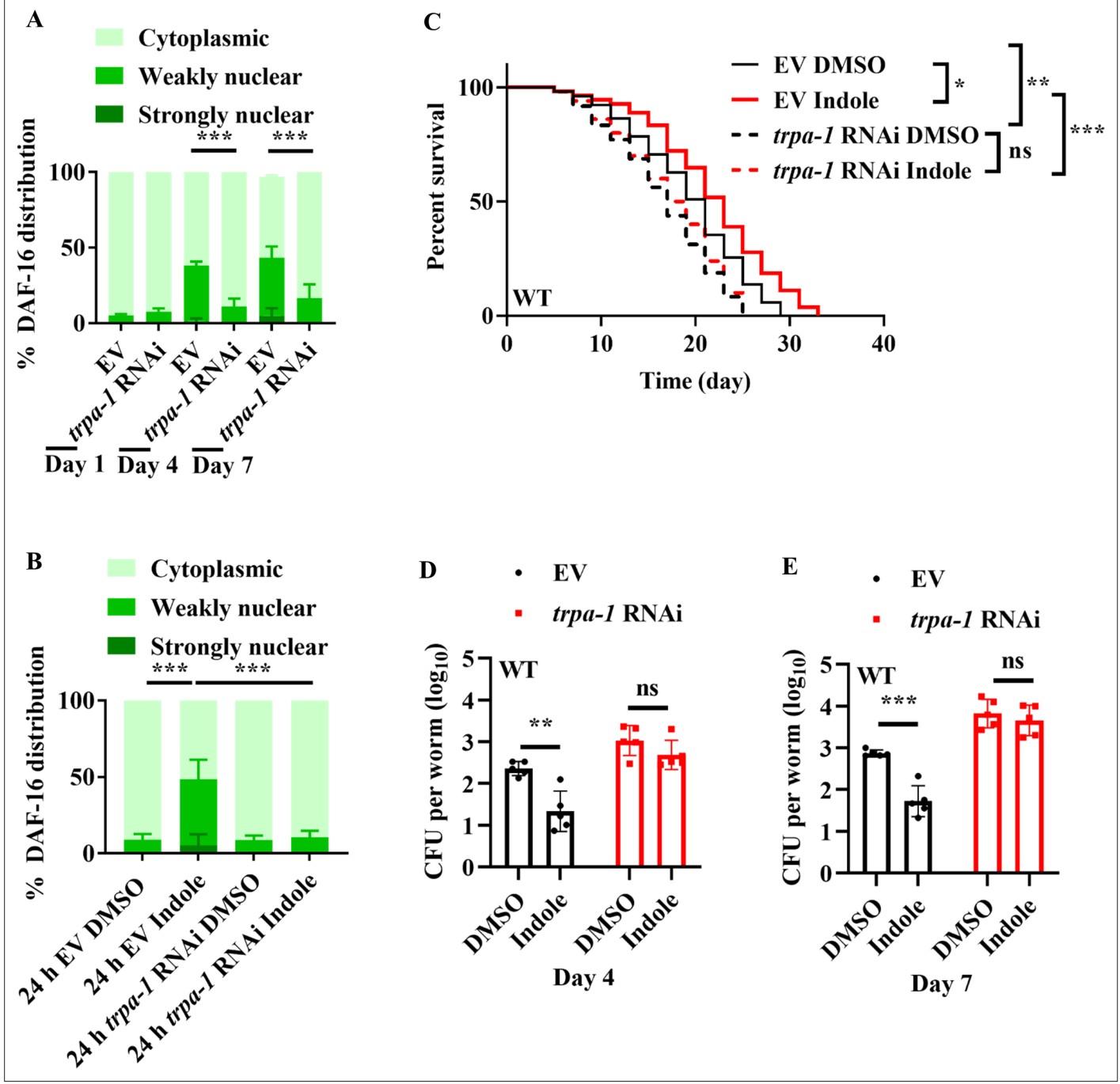

**Figure 4.** TRPA-1 is involved in indole-mediated DAF-16 nuclear translocation. Knockdown of *trpa-1* by RNAi suppressed the nuclear translocation of DAF-16::GFP in worms on days 4 and 7 (**A**), or in young adult worms treated with indole (100 μM) for 24 hr (**B**). EV, empty vector. These results are means ± standard deviation (SD) of three independent experiments (*n* > 35 worms per experiment). ***p < 0.001. ns, not significant. p-values (**A, B**) were calculated using the Chi-square test. (**C**) Knockdown of *trpa-1* by RNAi significantly shortened the lifespan of worms treated with indole (100 μM). *p < 0.05; **p < 0.01; ***p <0 .001. (**D, E**) Colony-forming units (CFU) of *E. coli* K12 were significantly increased in worms subjected to *trpa-1* RNAi on days 4 (**E**) and 7 (**F**). Meanwhile, supplementation with indole (100 μM) failed to suppress the increase in CFU in *trpa-1* (RNAi) worms. These results are means ± SD of five independent experiments (*n* > 30 worms per experiment). **p < 0.01; ***p < 0.001. ns, not significant. p-values (**D, E**) were calculated using a two-tailed *t*-test.

The online version of this article includes the following source data and figure supplement(s) for figure 4:

**Source data 1.** Lifespan assays summary and quantification results.

**Figure supplement 1.** Indole promotes nuclear localization of DAF-16 independent of AHR-1 and SGK-1.

*Figure 4 continued on next page*

*Figure 4 continued*

**Figure supplement 1—source data 1.** Quantification results.

**Figure supplement 2.** Indole extends lifespan and inhibits bacterial accumulation via TRPA-1–DAF-16 axis.

**Figure supplement 2—source data 1.** Lifespan assays summary and quantification results.

**Figure supplement 3.** Supplementation with indole fails to induce nuclear localization of SKN-1 in worms.

**Figure supplement 3—source data 1.** Quantification results.

Likewise, knockdown of *trpa-1* in either neurons (**Figure 5C, D**) or the intestine (**Figure 5E, F**) increased the CFU of *E. coli* K-12 in worms on days 4 and 7. However, supplementation with indole failed to inhibit the increase in the CFU of *E. coli* K-12 in worms subjected to neuronal-specific *trpa-1* RNAi (**Figure 5C, D**). In contrast, supplementation with indole significantly suppressed the CFU of *E. coli* K-12 in these worms subjected to intestinal-specific *trpa-1* RNAi (**Figure 5E, F**). These results suggest that TRPA-1 in neurons is involved in indole-mediated longevity.

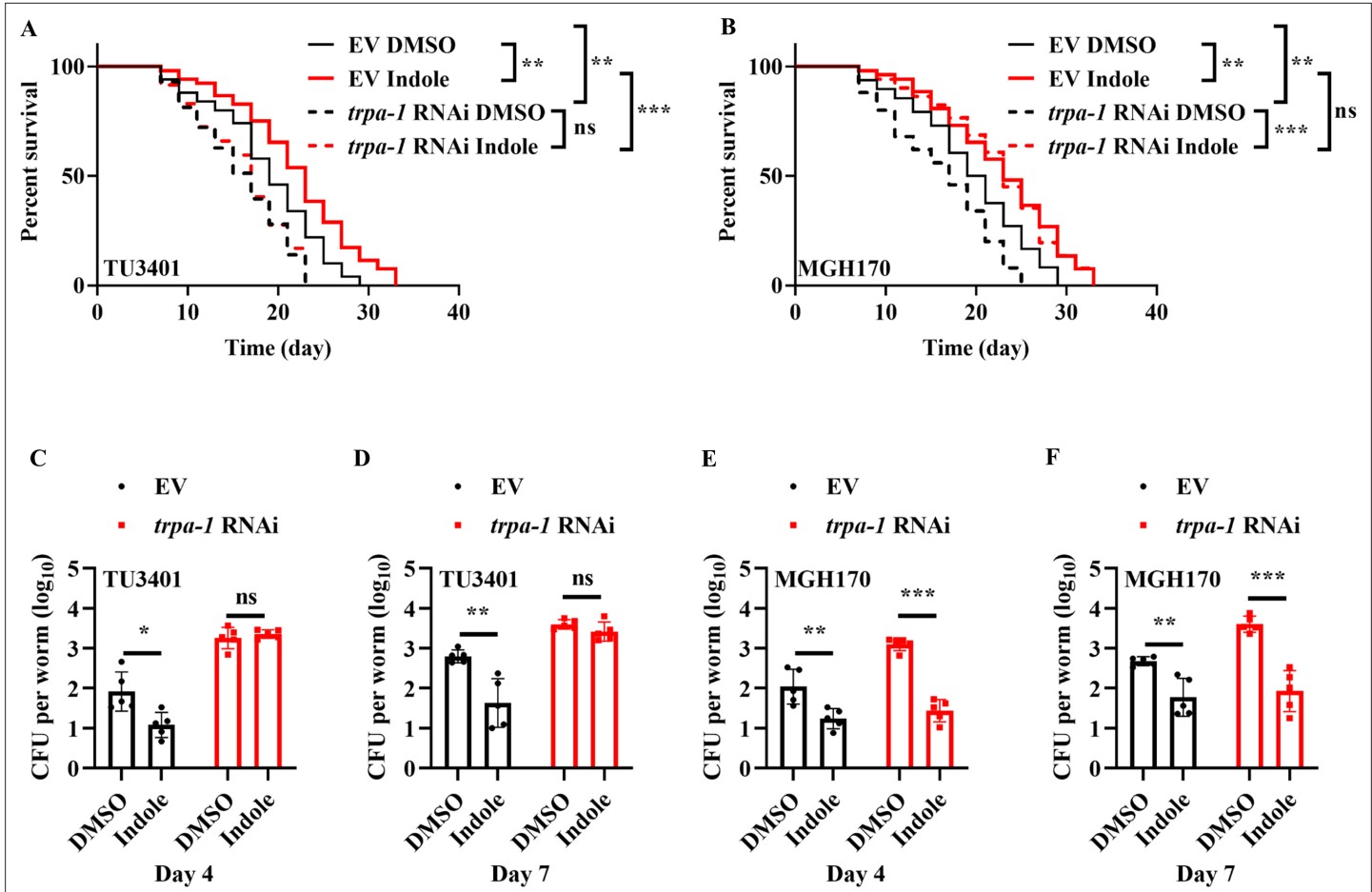

**Figure 5.** TRPA-1 in neurons is involved in indole-mediated longevity. Either neuronal- (**A**) or intestinal- (**B**) specific *trpa-1* RNAi shortened the lifespan of worms. However, supplementation with indole (100 µM) significantly extended the lifespan of worms after knockdown of *trpa-1* by RNAi in the intestine, but not in neurons. EV, empty vector. **p < 0.01; ***p < 0.001. ns, not significant. p-values (**A, B**) were calculated using log-rank test. Supplementation with indole (100 µM) no longer inhibited colony-forming units (CFU) of *E. coli* K12 in worms on days 4 (**C**) and 7 (**D**) after knockdown of *trpa-1* by RNAi in neurons. These results are means ± standard deviation (SD) of five independent experiments (*n* > 30 worms per experiment). *p < 0.05; **p < 0.01. ns, not significant. Supplementation with indole (100 µM) significantly suppressed the CFU of *E. coli* K-12 in worms subjected to intestinal-specific *trpa-1* RNAi on days 4 (**E**) and 7 (**F**). These results are means ± SD of five independent experiments (*n* > 30 worms per experiment). **p < 0.01; ***p < 0.001. ns, not significant. p-values (**C–F**) were calculated using a two-tailed *t*-test.

The online version of this article includes the following source data for figure 5:

**Source data 1.** Lifespan assays summary and quantification results.

## LYS-7 and LYS-8 functions as downstream molecules of DAF-16 to maintain normal lifespan in worms

*C. elegans* possesses a variety of putative antimicrobial effector proteins, such as lysozymes, defensin-like peptides, neuropeptide-like proteins, and caenacins (*Dierking et al., 2016*). Of these antimicrobial proteins, lysozymes are involved in host defense against various pathogens (*Boehnisch et al., 2011*; *Irazoqui et al., 2010a*; *Mallo et al., 2002*; *O'Rourke et al., 2006*; *Visvikis et al., 2014*). A previous study has demonstrated that expressions of lysozyme genes, such as *lys-2*, *lys-7*, and *lys-8*, are markedly up-regulated in 4-day-old worms, which is dependent on DAF-16 (*Li et al., 2019*). We thus determined the role of these lysozyme genes in the lifespan of worms. We found that a single

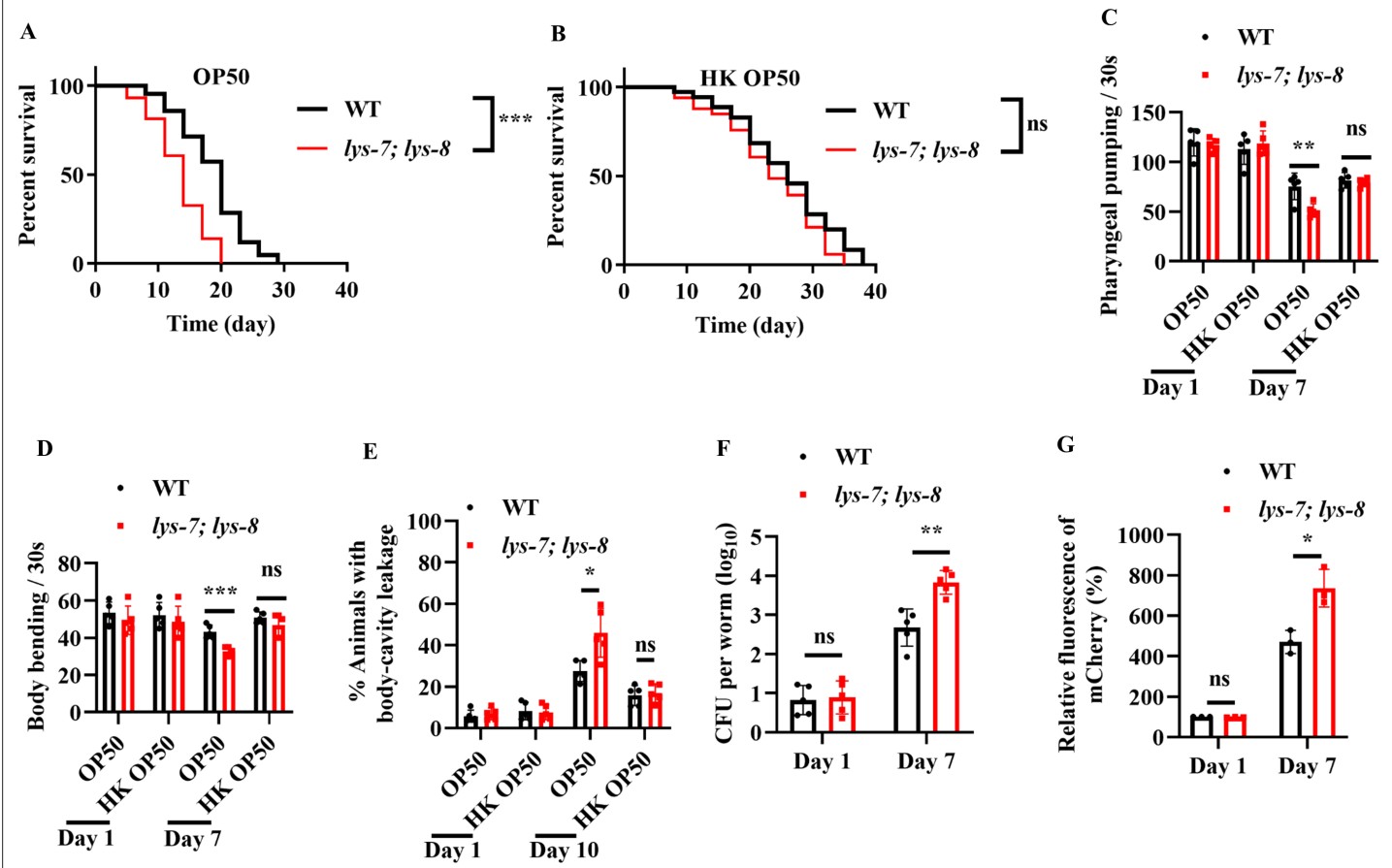

**Figure 6.** LYS-7 and LYS-8 are required for maintenance of normal lifespan in worms. The lifespans of worms fed either live *E. coli* OP50 (**A**) or heat-killed (HK) *E. coli* OP50 (**B**). The *lys-7(ok1384); lys-8(ok3504)* double mutants exhibited a shorter lifespan, than wild-type (WT) worms fed live *E. coli* OP50 (**A**). In contrast, the lifespan of the *lys-7(ok1384); lys-8(ok3504)* double mutants was comparable of that of WT worms fed HK *E. coli* OP50 (**B**). ***p < 0.001. ns, not significant. p-values (**A, B**) were calculated using log-rank test. (**C–E**) *lys-7* and *lys-8* were involved in delaying the appearance of the aging markers, including pharyngeal pumping (**C**), body bending (**D**), and body-cavity leakage (**E**) in worms fed live *E. coli* OP50. These results are means ± standard deviation (SD) of five independent experiments (*n* > 20 worms per experiment). *p < 0.05; **p < 0.01; ***p < 0.001. (**F**) Colony-forming units (CFU) of *E. coli* OP50 were significantly increased in *lys-7(ok1384); lys-8(ok3504)* double mutants on day 7. These results are means ± SD of five independent experiments (*n* > 20 worms per experiment). **p < 0.01. ns, not significant. (**G**) Quantification of fluorescent intensity of *E. coli* OP50 expressing mCherry in *lys-7(ok1384);lys-8(ok3504)* double mutants. These results are means ± SD of three independent experiments (*n* > 35 worms per experiment). *p < 0.05. ns, not significant. p-values (**C–G**) were calculated using a two-tailed *t*-test.

The online version of this article includes the following source data and figure supplement(s) for figure 6:

**Source data 1.** Lifespan assays summary and quantification results.

**Figure supplement 1.** The roles of lysozyme genes in lifespan in worms.

**Figure supplement 1—source data 1.** Lifespan assays summary.

**Figure supplement 2.** Indole-mediated lifespan extension in worms depends on LYS-7 and LYS-8.

**Figure supplement 2—source data 1.** Lifespan assays summary and quantification results.

mutation in *lys-7(ok1384)* slightly but significantly reduced the lifespan of worms fed live *E. coli* OP50 (*Figure 6—figure supplement 1A*), but not HK *E. coli* OP50 (*Figure 6—figure supplement 1B*), which was consistent with a previous observation (*Portal-Celhay et al., 2012*). In contrast, whereas either a single mutation in *lys-8(ok3504)* or RNAi knockdown of *lys-2* did not affect the lifespan when worms were grown on live *E. coli* OP50, or HK *E. coli* OP50 (*Figure 6—figure supplement 1A, B*). However, *lys-7(ok1384); lys-8(ok3504)* double mutants exhibited a shortened lifespan when worms were fed live *E. coli* OP50 (*Figure 6A*). In contrast, the lifespan of *lys-7(ok1384); lys-8(ok3504)* double mutants was comparable to that of WT worms fed HK *E. coli* OP50 (*Figure 6B*). Furthermore, knock-down of *lys-2* by RNAi did not affect the lifespan of *lys-7(ok1384); lys-8(ok3504)* double mutants grown on live *E. coli* OP50, or HK *E. coli* OP50 (*Figure 6—figure supplement 1C, D*). Both the rates of pharyngeal-pumping (*Figure 6C*) and body bending (*Figure 6D*) were reduced in 8-day-old *lys-7(ok1384); lys-8(ok3504)* double mutants fed live *E. coli* OP50. Moreover, the body-cavity leakage was increased in the double mutants fed live *E. coli* OP50 on day 10 (*Figure 6E*). However, these age-associated markers in *lys-7(ok1384); lys-8(ok3504)* double mutants were comparable to those in age-matched WT worms fed HK *E. coli* OP50. In addition, double mutations in *lys-7(ok1384); lys-8(ok3504)* increased the CFU of *E. coli* OP50 (*Figure 6F*) as well as the accumulation of *E. coli* OP50 expressing RFP in worms on day 7 (*Figure 6G*). Finally, we found that supplementation with indole no longer extended the lifespan of worms and failed to suppress the increase the CFU of *E. coli* K-12 in *lys-7(ok1384); lys-8(ok3504)* double mutants (*Figure 6—figure supplement 2A, B*).

Using the transgenic worms expressing either *lys-7p::gfp* or *lys-8p::gfp*, we found that the expressions of *lys-7p::gfp* and *lys-8p::gfp* were significantly up-regulated in worms fed live *E. coli* OP50 on days 4 and 7 (*Figure 7—figure supplement 1A–D*), but not in age-matched worms fed HK *E. coli* OP50 (*Figure 7—figure supplement 1A–D*). RNAi knockdown of *daf-16* also significantly suppressed the expressions of *lys-7p::gfp* and *lys-8p::gfp* in these worms fed live *E. coli* OP50 (*Figure 7A, B*). Similar results were obtained by measuring the mRNA levels of *lys-7* and *lys-8* in *daf-16(mu86)* mutants using quantitive real-time PCR (qPCR) (*Figure 7—figure supplement 1E, F*). Furthermore, we found that expression of either *lys-7p::gfp* (*Figure 7C*) or *lys-8p::gfp* (*Figure 7D*) was reduced in worms fed *E. coli* K-12 *ΔtnaA* strain on days 4 and 7, compared to those in age-matched worms fed *E. coli* K-12 strain. Likewise, indole (100 µM) remarkably increased the expression of either *lys-7p::gfp* or *lys-8p::gfp* in young adult worms after 24 hr treatment (*Figure 7E, F*). Meanwhile, RNAi knockdown of *trpa-1* significantly inhibited the expression of either *lys-7p::gfp* or *lys-8p::gfp* in 4-day-old worms fed *E. coli* K12 strain (*Figure 7A, B*). Finally, we found that the mRNA levels of *lys-7* and *lys-8* were significantly down-regulated in worms subjected to neuronal specific (*Figure 7—figure supplement 2A*), but not intestinal specific, knockdown of *trpa-1* by RNAi on day 4 (*Figure 7—figure supplement 2B*). However, supplementation with indole only up-regulated the expression of *lys-7* and *lys-8* in worms subjected to intestinal specific (*Figure 7—figure supplement 2C*), but not neuronal specific, RNAi of *trpa-1* (*Figure 7—figure supplement 2D*). These results suggest that LYS-7 and LYS-8 function as downstream molecules of DAF-16 to maintain normal lifespan by inhibiting bacterial proliferation in worms.

## Discussion

Using *C. elegans* and its dietary bacterium as a host–microbe model, we provide a striking example of how a host responds to microbial dysbiosis in the gut (*Figure 8*). As worms age, *E. coli* proliferates in the lumen, thereby producing and secreting more indole. When its concentration crosses a threshold value, the bacterially produced compound is perceived by TRPA-1 in worms. The TRPA-1 signaling triggers DAF-16 nuclear translocation, leading to up-regulation of lysozyme genes. These antimicrobial peptides help worms to maintain normal lifespan by limiting the bacterial proliferation.

A complication in understanding the impact of bacterial diets on the traits of the worm is the fact that initially bacteria are a source of food, but later become pathogenic (*Garigan et al., 2002*; *Tan and Shapira, 2011*). Thus, the accumulation of *E. coli* in the gut is harmful for organismal fitness of worms during the course of life. As a well-known regulator of longevity and innate immunity (*Garsin et al., 2003*; *Zou et al., 2013*), DAF-16 is activated in aged *C. elegans* (*Li et al., 2019*). This transcription factor is involved in both maintaining normal lifespan and limiting proliferation of *E. coli* in worms (*Garigan et al., 2002*; *Portal-Celhay and Blaser, 2012*; *Portal-Celhay et al., 2012*). In this study, our data demonstrate that activation of DAF-16 requires contact with live bacterial cells in the gut of

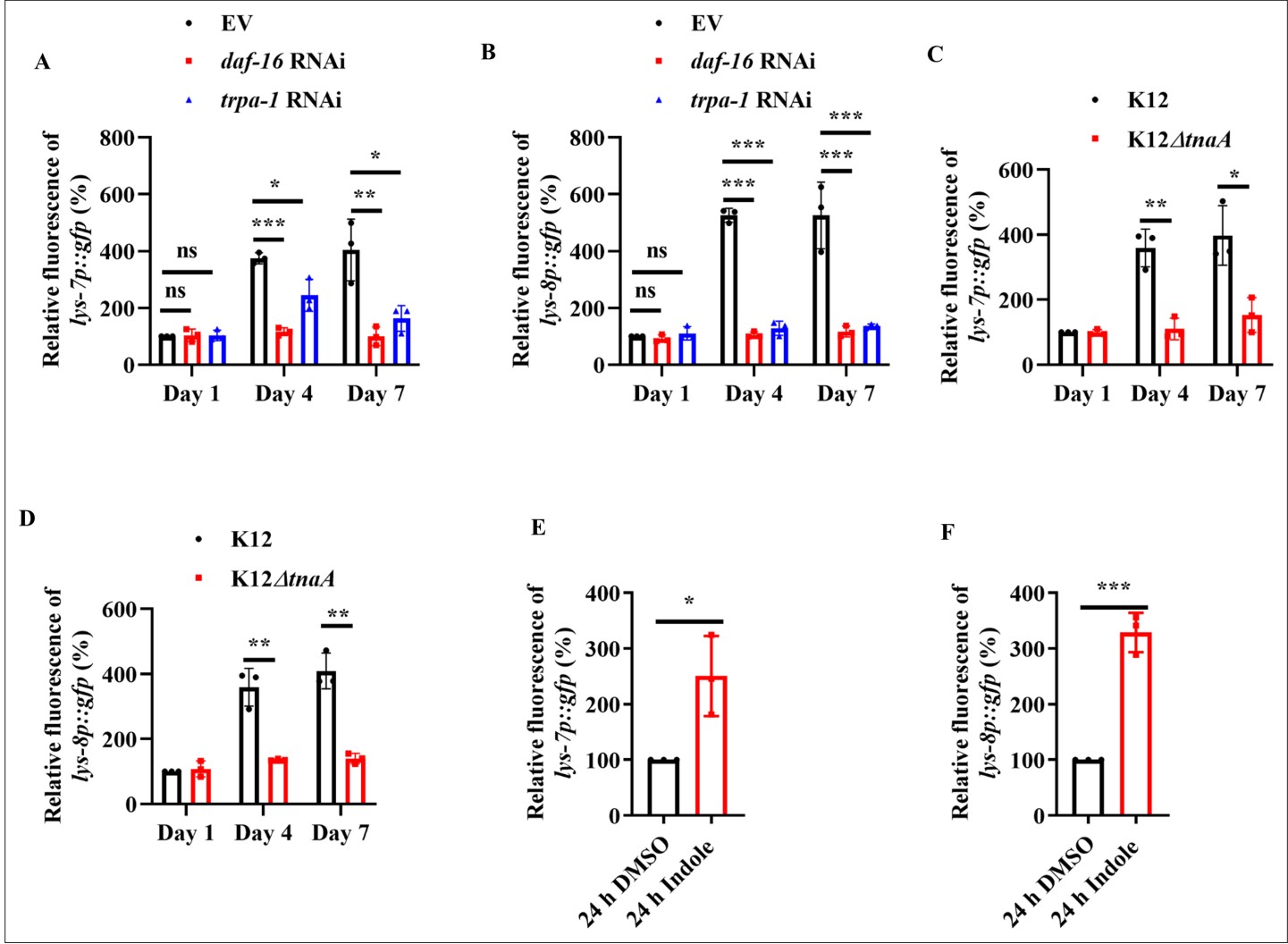

**Figure 7.** The expressions of *lys-7* and *lys-8* were up-regulated by the indole/TRPA-1/DAF-16 signaling. The expression of either *lys-7p::gfp* (**A**) or *lys-8p::gfp* (**B**) was significantly suppressed after knockdown of *daf-16* or *trpa-1* by RNAi in worms fed live *E. coli* OP50 on days 4 and 7. EV, empty vector. The expression of either *lys-7p::gfp* (**C**) or *lys-8p::gfp* (**D**) was reduced in worms fed *E. coli* K-12 *ΔtnaA* strain on days 4 and 7, compared with that in age-matched worms fed *E. coli* K-12. Indole (100 μM) remarkably increased the expression of either *lys-7p::gfp* (**E**) or *lys-8::gfp* (**F**) in young adult worms after 24 hr of treatment. These results are means ± standard deviation (SD) of three independent experiments (*n* > 35 worms per experiment). *p < 0.05; **p < 0.01; ***p < 0.001. ns, not significant. p-values (**A–F**) were calculated using a two-tailed *t*-test.

The online version of this article includes the following source data and figure supplement(s) for figure 7:

**Source data 1.** Quantification results.

**Figure supplement 1.** The expressions of *lys-7p::gfp* and *lys-8p::gfp* are up-regulated in worms with age.

**Figure supplement 1—source data 1.** Quantification results.

**Figure supplement 2.** TRPA-1 in neurons is required for the expression *lys-7* and *lys-8*.

**Figure supplement 2—source data 1.** Quantification results.

worms as dead *E. coli* fails to activate DAF-16. Thus, the accumulation of *E. coli* during aging, but not aging itself, results in the activation of DAF-16. Furthermore, DAF-16 mutation does not influence the lifespan in worms fed dead *E. coli*. Taken together, these findings clearly demonstrate that DAF-16 acts to maintain homeostasis by inhibiting bacterial proliferation in worms with age.

Indole is produced from tryptophan by tryptophanase in a large number of bacterial species. As a well-known signaling molecule, indole is involved in regulation of a variety of physiological processes in bacteria, such as cell division, biofilm formation, virulence, spore formation, and antibiotic resistance (*Lee et al., 2015*; *Zarkan et al., 2020*). Although animals cannot synthesize indole, they can sense and

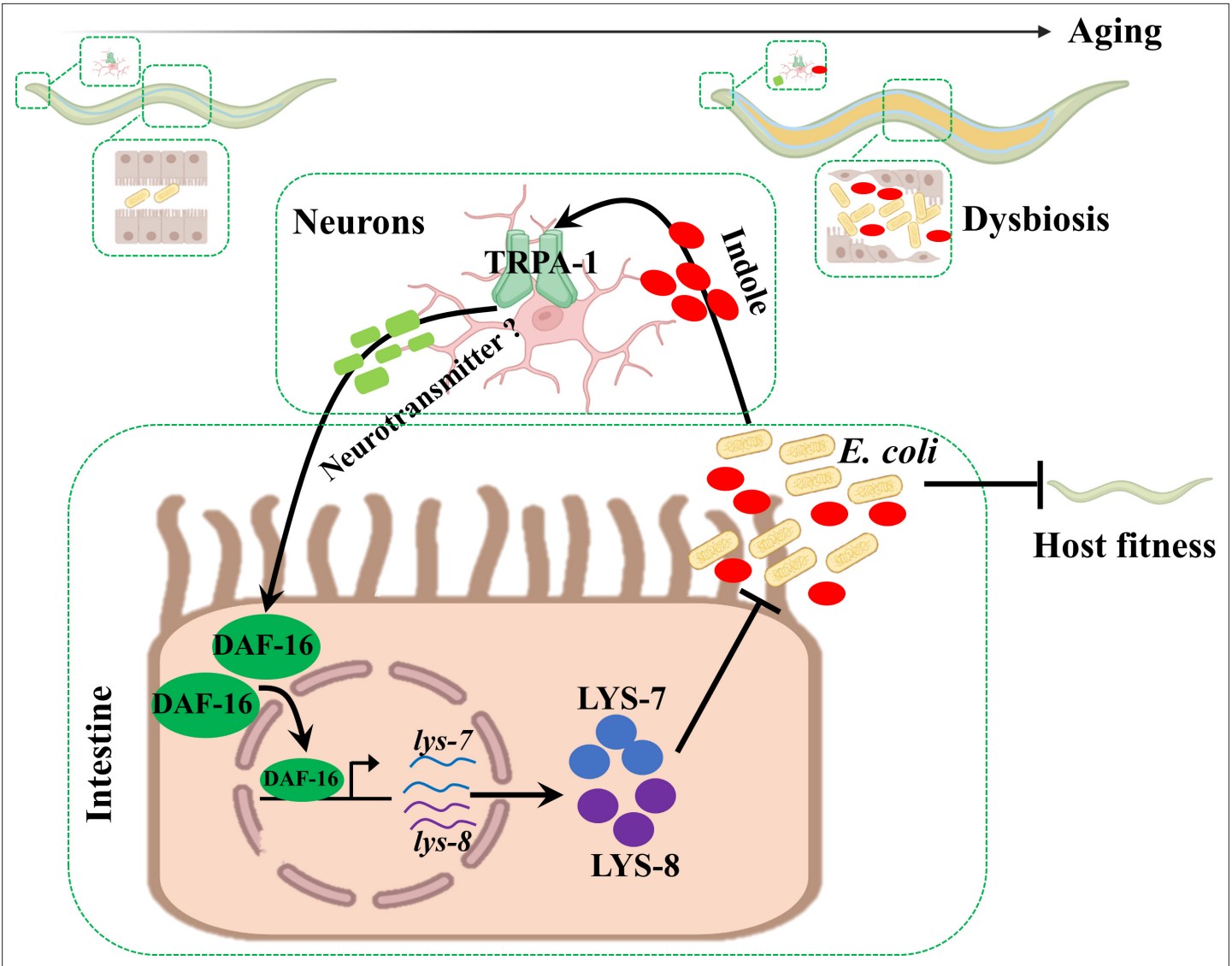

**Figure 8.** Schematic model of indole as a microbial signal of gut dysbiosis in maintaining organismal fitness. This study demonstrates that intestinal accumulation of *E. coli* in *C. elegans* with age leads to elevated levels of indole, which activates FOXO/DAF-16 transcription factor in the intestine via a cold-sensitive TRP channel TRPA-1 in neurons. The transcription factor in turn up-regulates the expression of lysozyme genes, thereby limiting the bacterial overproliferation in the gut to support organismal fitness in worms.

modify this metabolite (*Lee et al., 2015*). Indole and its derivatives can influence insect behaviors and human diseases, such as intestinal inflammation and diabetes (*Agus et al., 2018*; *Lee et al., 2015*). Thus, indole may function as an interspecies and interkingdom signaling molecule to influence the microbe–host interaction (*Bansal et al., 2010*; *Oh et al., 2012*; *Ye et al., 2021*). For instance, enteric delivery of indole (1 mM) increases the intestinal motility by inducing 5-hydroxytryptamine secretion in zebrafish larvae (*Ye et al., 2021*). Supplementation with 0.2 mM indole enhances the resistance of *C. elegans* to infection with *C. albicans* by reducing fungal colonization in the intestine (*Oh et al., 2012*). In addition, treatment of HCT-8 intestinal epithelial cells with 1 mM indole inhibits tumor necrosis factor α (TNF-α) mediated activation of nuclear factor-kappa B (NF-κB) and up-regulation of the inflammatory factor IL-8, thus improving intestinal epithelial barrier function (*Bansal et al., 2010*). Our data show that indole limits the bacterial proliferation in the gut of worms by driving intestinal defense gene expression via DAF-16. These findings suggest that the bacteria-derived metabolite may serve as a pathogen-associated molecular pattern that is recognized by metazoans. Although indole at a higher concentration (5 mM) is capable of inhibiting cell division in *E. coli* K12 strain (*Chimerel et al.,*

*2012*), exogenous indole has little effect on the growth of *E. coli* K12 strain up to 3 mM (*Chant and Summers, 2007*). In general, extracellular indole concentrations detected in stationary phase LB cultures are typically 0.5–1 mM depending on the specific *E. coli* strain (*Zarkan et al., 2020*). Our data show that indole is sufficient to inhibit the accumulation of *E. coli* BW25113 at the concentration range from 0.05 to 0.2 mM. Thus, these results suggest that direct causal involvement of cell cycle arrest in *E. coli* by indole is unlikely.

Our data demonstrate that the cold-sensitive TRP channel TRPA-1 in neurons is involved in indole-mediated nuclear translocation of DAF-16 in the intestine, which is required for lifespan extension in *C. elegans*. Recently, *Ye et al., 2021* have shown that both indole and indole-3-carboxaldehyde (IAld) are TRPA-1 agonists in vertebrates. *trpa-1* is widely expressed in a variety of sensory neurons in worms (*Kindt et al., 2007*). Low temperature also activates TRPA-1, which in turn acts to promote longevity via the PKC-2–SGK-1–DAF-16 pathway (*Xiao et al., 2013*). It should be noted that unlike indole, activation of TPRA-1 by low temperature promotes the transcription activity, but not the nuclear translocation, of DAF-16 (*Xiao et al., 2013*). Furthermore, our data show that knockdown of *sgk-1* dose not influence the nuclear translocation of DAF-16 induced by indole. Finally, indole extends lifespan via TRPA-1 only in nervous system, whereas low temperature extends lifespan via TRPA-1 both in the intestine and neurons. Thus, indole-mediated lifespan extension is essentially different from low temperature-mediated lifespan extension, although TRPA-1 and DAF-16 are involved in both processes. One possibility is that only neurons with activated TRPA-1 may release a neurotransmitter, which in turn triggers a signaling pathway to extend the lifespan of worms via activating DAF-16 in a non-cell autonomous manner, while the intestinal activated TRPA-1 is unable to release the neurotransmitter. Indeed, TRPA-1 induces the releasing of calcitonin gene-related peptide in perivascular sensory nerves, which in turn binds corresponding G-protein-coupled receptor, and causes membrane hyperpolarization and arterial dilation on smooth muscle cells (*Talavera et al., 2020*). Recently, Ye et al. have shown that both indole and indole-3-carboxaldehyde (IAld) are TRPA-1 agonists in vertebrates (*Ye et al., 2021*). As an agonist of TRPA-1, podocarpic acid can activate SKN-1 via TRPA-1 in worms (*Chaudhuri et al., 2016*). However, podocarpic acid fails to trigger DAF-16 translocation. Unlike podocarpic acid, indole cannot activate SKN-1 in worms. Although both indole and podocarpic acid are non-electrophilic agonists of TAPR-1, the structures of podocarpic acid and indole are totally different. These results suggest that TRPA-1 activation by podocarpic acid, low temperature, and indole occurs largely through distinct mechanisms.

In this study, our data demonstrate that DAF-16 inhibits bacterial proliferation by up-regulating expression of two lysozyme genes, *lys-7* and *lys-8*. As a group of digestive enzymes with antimicrobial properties, lysozymes play an important role in the innate immunity in both vertebrate and invertebrate animals (*O'Rourke et al., 2006*). As a target gene of DAF-16, *lys-7* has been proven to play an important role in resistance against a variety of pathogens, such as *Microbacterium nematophilum* (*O'Rourke et al., 2006*), *Pseudomonas aeruginosa* (*Nandakumar and Tan, 2008*), the pathogenic *E. coli* LF82 (*Simonsen et al., 2011*), *Bacillus thuringiensis* (*Boehnisch et al., 2011*), and *Cryptococcus neoformans* (*Marsh et al., 2011*). A previous study has demonstrated that a mutation in *lys-7* significantly reduces the lifespan, but does not influence the accumulation of *E. coli* OP50 in the intestine of 2-day-old worms (*Portal-Celhay et al., 2012*). In the current study, the *lys-7* mutants exhibit reduced the lifespan and elevated bacterial loads in the intestine of worms on days 4 and 7. This discrepancy may be due to the worms at different ages. Actually, our data also show that the *lys-7;lys-8* double mutants exhibit a comparable accumulation of *E. coli* OP50 in worms on day 1. Although *lys-8* mutation dose not influence either the lifespan or bacterial loads, it enhances the effect of *lys-7*. These results suggest that *lys-8* acts in synergy with *lys-7* to limit bacterial accumulation in the gut of worms.

It has been well established that bacterial dysbiosis is significantly associated with IBDs (*Manichanh et al., 2012*; *Sommer and Bäckhed, 2013*). Interestingly, reduced levels of indole and its derivative indole-3-propionic acid (IPA) are observed in serum of mice with dextran sulfate sodium-induced colitis and patients with IBD (*Alexeev et al., 2018*). Oral administration of IPA significantly ameliorates disease symptoms and promotes intestinal homeostasis by up-regulating colonic epithelial IL-10R1 in the chemically induced murine colitis model. Thus, characterization of the role for indole and its derivatives in host–microbiota interactions within the mucosa may provide new therapeutic avenues for inflammatory intestinal diseases.

## Materials and methods

### Nematode strains

*daf-16(mu86)*, *lys-7(ok1384)*, *lys-8(ok3504)*, LD1*[skn-1b/c::gfp+rol-6(su1006)]*, the nematode strain for neuronal-specific RNAi, TU3401 (*sid-1(pk3321); uIs69 [pCFJ90 (myo-2p:: mCherry)+unc-119p::sid-1]*), TJ356 (*zIs356 [daf-16p::daf-16a/b::GFP+rol-6 (su1006)]*), and SAL105 (*denEx2 [lys-7::GFP+pha-1(+)]*) were kindly provided by the *Caenorhabditis* Genetics Center (CGC; http://www.cbs.umn.edu/CGC), funded by NIH Office of Research Infrastructure Programs (P40 OD010440). The nematode strain for intestinal-specific RNAi, MGH170 (*sid-1(qt9); Is[vha-6pr::sid-1]; Is[sur-5pr::GFPNLS]*), was kindly provided by Dr. Gary Ruvkun (Massachusetts General Hospital, Harvard Medical School). The strain MQD1586 (*476[hsp-16.2p::nCherry; dod-3p::gfp; mtl-1::bfp, unc-119(+)]*) was kindly provided by Dr. Mengqiu Dong (Beijing Institute of Life Sciences). The strain *trpa-1(ok999)* was kindly provided by Dr. Jianke Gong (Huazhong University of Science and Technology). Mutants were backcrossed three times into the N2 strain used in the laboratory. All strains were maintained on NGM and fed with *E. coli* OP50 at 20°C.

### Study design for age-related dysbiosis study

Synchronized L1 larvae were grown on NGM agar plates seeded with *E. coli* OP50 at 20°C until they reached the young adult stage. All the experiments started from the young adult stage, which was considered day 0. From days 1 to 10, the worms were transferred to new NGM plates containing *E. coli* strains at 20°C daily for further experiments. For preparing for HK *E. coli* OP50, the bacterium cultured overnight in liquid LB medium was treated with 75°C water bath for 90 min (*Qi et al., 2017*). For preparing for ampicillin-killed *E. coli* OP50, the bacterium cultured overnight in liquid LB medium was seeded onto NGM plates containing 200 μg/ml ampicillin. Then the OP50 strain was periodically streaked onto ampicillin-containing and ampicillin-free LB plates to confirm that it could not proliferate (*Garigan et al., 2002*). BW25113 (*E. coli* K-12 WT) and *E. coli* K-12 Δ*tnaA* strain were obtained from the Keio collection (*Baba et al., 2006*).

### RNA interference

RNAi bacterial strains containing targeting genes were obtained from the Ahringer RNAi library (*Kamath and Ahringer, 2003*). All clones used in this study were verified by sequencing. Briefly, *E. coli* strain HT115 (DE3) expressing dsRNA was grown in LB (Luria-Bertani) containing 100 μg/ml ampicillin at 37°C for overnight, and then spread onto NGM plates containing 100 μg/ml ampicillin and 5 mM isopropyl 1-thio-β-D-galactopyranoside. The RNAi-expressing bacteria were then grown at 25°C overnight. Synchronized L1 larvae were placed on the plates at 20°C until they reached maturity. Young adult worms were used for further experiments.

### Construction of transgenic strains

The vector expressing *lys-8p::gfp* was generated by subcloning a 2011-bp promoter fragment of *lys-8* into an expression vector (pPD95.75). The vector was injected into the syncytial gonads of WT worms with 50 ng/ml pRF4 as a transformation marker (*Mello and Fire, 1995*). The transgenic worms carrying were confirmed before assay.

### DAF-16 nuclear localization assay

For the effect of aging on DAF-16::GFP localization, worms expressing *daf-16p::daf-16::gfp* were cultured on standard NGM plates at 20°C for 1, 4, and 7 days, respectively. For indole treatment, young adults were transferred to NGM plates containing 50–200 μM indole (Macklin, Shanghai, China) dissolved in dimethylsulfoxide (DMSO) for 24 hr at 20°C. NGM plates with equal amount of DMSO served as a control. After taken from incubation, worms were immediately mounted in M9 onto microscope slides. The slides were viewed using a Zeiss Axioskop 2 plus fluorescence microscope (Carl Zeiss, Jena, Germany) with a digit camera. The status of DAF-16 localization was categorized as cytosolic localization, nuclear localization when localization is observed throughout the entire body, or intermediate localization when nuclear localization is visible, but not completely throughout the body (*Oh et al., 2005*). At least 35 nematodes were counted per assay in three independent experiments.

## Fluorescence microscopic analysis

For imaging fluorescence, worms expressing *hsp-16.2p::nCherry*, *dod-3p::gfp*, *lys-7p::gfp*, and *lys-8p::gfp* were mounted in M9 onto microscope slides. The slides were imaged using a Zeiss Axioskop 2 Plus fluorescence microscope. The intensities of nCherry and GFP were analyzed using the ImageJ software (NIH). Three plates of at least 35 animals per plate were tested per assay, and all experiments were performed three times independently.

## Lifespan analysis

Synchronized L1 larvae were grown on NGM agar plates seeded with *E. coli* OP50 at 20°C until they reached the young adult stage. All the lifespan assays started from the young adult stage at 20°C. The first day of adulthood was recorded as day 1. From days 1 to 10, the worms were transferred to new NGM plates containing *E. coli* strains at 20°C daily. After that, worms were transferred every third day. The number of worms was counted every day. Worms that did not move when gently prodded and displayed no pharyngeal pumping were marked as dead. Every lifespan detection had three independent experiments (*n* > 35 worms per experiment).

## Age-related phenotypic marker assays

The following two age-related phenotypes were scored in 1- and 7-day-old worms (*Chen et al., 2019*). Pharyngeal pumping was measured by counting the number of contractions in the terminal bulb of pharynx in 30-s intervals. Body bending was measured by counting the number of body bends in 30-s intervals. At least 20 animals were determined per assay in five independent experiments.

## Intestinal barrier function assay in worms

Intestinal barrier function was determined according to the method described previously (*Ma et al., 2020*). Briefly, synchronized young adult animals were cultured on standard NGM plates at 20°C for 1, 4, 7, and 10 days. After removed from the NGM plates, these animals were suspended in M9 liquid medium containing *E. coli* OP50 (OD = 0.5–0.6), 5% food dye FD&C Blue No. 1 (Bis[4-(*N*-ethyl-*N*-3-sulfophenylmethyl) aminophenyl]-2-sulfophenylmethylium disodium salt) (AccuStandard, New Haven, CT), and incubated for 6 hr. After collected and washed with M9 buffer four times, the worms were mounted in M9 onto microscope slides. The slides were viewed using a Zeiss Axioskop 2 Plus fluorescence microscope (Carl Zeiss, Jena, Germany) to measure the leakage of the dyes in the body cavity of animals. The rate of body-cavity leakage was calculated as a percentage by dividing the number of animals with dye leakage by the number of total animals. For each time point, five independent experiments were carried out. In each experiment, at least 20 of worms were calculated.

## Detection of bacterial accumulation in worms

For detection of *E. coli* accumulation, worms were grown on NGM plates with *E. coli* expressing mCherry (the plasmid of PMF440 purchased from addgene) for 1, 4, and 7 days at 20°C. Then animals were collected and soaked in M9 buffer containing 25 mM levamisole hydrochloride (Sangon Biotech Co), 50 μg/ml kanamycin (Sangon Biotech Co), and 100 μg/ml ampicillin (Sangon Biotech Co) for 30 min at room temperature. Then worms were washed three times with M9 buffer. Some of animals were mounted in M9 onto microscope slides. The slides were viewed using a Zeiss Axioskop 2 plus fluorescence microscope (Carl Zeiss, Jena, Germany) with a digital camera. At least 35 worms were examined per assay in three independent experiments. Meanwhile, at least 30 of nematodes were transferred into 50 μl PBS plus 0.1% Triton and ground. The lysates were diluted by 10-fold serial dilutions in sterilized water and spread over LB agar plates with 100 μg/ml ampicillin. After incubation overnight at 37°C, the *E. coli* CFU was counted. For each group, three to five independent experiments were carried out.

## Isolation and identification of the active compound

Ten liters of the *E. coli* OP50 culture supernatants were collected by centrifugation, and freeze-dried in a VirTis freeze dryer. The powders were then dissolved with 10 ml of methanol. The crude extract was loaded on to an Ultimate 3000 HPLC (Thermo Fisher, Waltham, MA) coupled with automated fraction collector in batches through a continuous gradient on an Agilent ZORBAX SB-C18 column (Agilent, 5 μm, 4.6 × 250 mm) at a column temperature of 40°C to yield 45 fractions based on retention time

and one fraction was collected per minute. The total flow rate was 1 ml/min; mobile phase A was 0.1% formic acid in water; and mobile phase B was 0.1% formic acid in acetonitrile. The HPLC conditions were manually optimized on the basis of separation patterns with the following gradient: 0–2 min, 10% B; 10 min, 25% B; 30 min, 35% B; 35 min, 90% B; 36 min, 95% B; 40 min, 90% B; 40.1 min, 10% B; and 45 min, 10% B. UV spectra were recorded at 204–400 nm. The injection volume for the extracts was 50 μl. After freeze-drying each fraction, 1 mg metabolites were dissolved in DMSO for DAF-16 nuclear localization assay (*Supplementary file 2*). The 26th fraction with DAF-16 nuclear translocation-inducing activity was further separated on silica gel column (200–300 mesh) with a continuous gradient of decreasing polarity (100%, 70%, 50%, 30%, petroleum ether/acetone) to yield four fractions (26a, 26b, 26c, and 26d). The fraction 26b with inducing DAF-16 nuclear translocation-inducing activity was further purified using a Sephadex LH-20 column to yield 32 fractions. The fraction 26b-11 with DAF-16 nuclear translocation-inducing activity contained a high-purity compound through thin layer chromatography detection. Then 26b-11 was structurally elucidated with NMR and MS data. NMR experiments were carried out on a Bruker DRX-500 spectrometer (Bruker Corp, Madison, WI) with solvent as internal standard. High-resolution MS data were performed on Q Exactive Focus UPLC-MS (Thermo Fisher) with a PDA detector and an Obitrap mass detector using positive mode electrospray ionization.

## Quantitative analysis of indole in worms

For quantitation of indole in *C. elegans*, worms were collected, washed three to five times with M9 buffer, and lyophilized for 4–6 hr using a VirTis freeze dryer. Dried pellets were weighed, and transferred to a 1.5-ml centrifuge tube. After grinded for 10 min in a tissue grinder, the samples were dissolved in 300 μl solvent (methanol:water, 20:80 vol/vol). The mixtures were then grinded for another 10 min, and centrifuged at $12,000 \times g$ for 5 min. The supernatants were collected, and filtered through 0.22 μm membranes for further analysis using LC–MS. LC–MS analyses were performed on a Q Exactive Focus UPLC-MS (Thermo Fisher) with an atmospheric pressure chemical ionization (APCI) source and operated with positive mode and coupled with Atlabtis dC18 column (Waters, 3 μm, 2.1 × 150 mm). Five microliters of samples were injected to the LC–MS system for analysis. The total flow rate was 0.3 ml/min; mobile phase A was 0.1% formic acid in water; and mobile phase B was 0.1% formic acid in acetonitrile. A/B gradient started at 5% B for 3 min after injection and increased linearly to 95% B at 13 min, then back to 5% B over 0.1 min and finally held at 5% B for an additional 1.9 min to re-equilibrate the column. Quantitation of indole was then achieved using standard curves generated using indole standard. Standard curve concentrations at 10, 50, 100, 300, 500, 700, and 9000 pmol/l in the solvent (methanol:water, 20:80 vol/vol). The data were analyzed and processed using Xcalibur software (Thermo Fisher).

## Construction of K12△*tnaA::tnaA* strain

All the PCR products were amplified using GXL high-fidelity DNA Polymerase (TaKaRa Biotechnology Co Ltd, Dalian, China). A 1846-bp fragment encompassing the *tnaA* gene transcription unit was amplified from the genomic DNA of *E. coli* K12 (*Abdelaal and Yazdani, 2021*). The transcription unit fragment and vector fragment were ligated by In-fusion Kit. The products were transformed into DH5α competent cells. Then the cells were harvested, and spread onto selective medium containing 50 μg/ml of aburamycin. The plate was incubated overnight at 37°C until colonies were observed. After transformed colonies were screened by colony PCR, the correct positive transformant was transferred into an LB liquid medium (50 μg/ml aburamycin), and incubated overnight. Positive plasmids were extracted, and verified by DNA sequencing. The sequenced *tnaA* rescued plasmids were transformed into *tnaA* strain by electroporation method. The *tnaA* rescued strains were cultured for contents and DAF-16 nuclear translocation assay.

## Quantitative real time PCR analysis

Total RNA from worms was isolated using Trizol reagent (Invitrogen, Carlsbad, CA). Random-primed cDNAs were generated by reverse transcription of the total RNA samples using a standard protocol. A quantitative real-time PCR analysis was performed with a Roche Light Cycler 480 System (Roche Applied Science, Penzberg, Germany) using SYBR Premix (Takara, Dalian, China). The relative amount of *lys-7* or *lys-8* mRNA to *act-1* mRNA (an internal control) was calculated using the method described

previously (*Pfaffl, 2001*). The primers used for PCR were as follows: *act-1*: 5′-CGT GTT CCC ATC CAT TGT CG-3′ (F), 5′-AAG GTG TGA TGC CAG ATC TTC-3′ (R); *lys-7*: 5′-GTC TCC AGA GCC AGA CAA TCC-3′ (F), 5′-CCA GTG ACT CCA CCG CTG TA-3′ (R); *lys-8*: 5′-GCT TCA GTC TCC GTC AAG GTC-3′ (F), 5′-TGA AGC TGG CTC AAT GAA AC-3′ (R).

## Statistics

Differences in survival rates were analyzed using the log-rank test. Differences in mRNA levels, the percentage of body-cavity leakage, fluorescence intensity, indole contents, and CFU were assessed by performing one-way ANOVA followed by the Student–Newman–Keuls test or by performing a two-tailed *t*-test. Differences in DAF-16::GFP nuclear accumulation were analyzed using the Chi-square test. Data were analyzed using GraphPad Prism 8.0.

## Acknowledgements

We thank the *Caenorhabditis* Genetics Center, Dr. Gary Ruvkun, Dr. Mengqiu Dong, and Dr. Jianke Gong for nematode strains. This work was supported by a grant from the National Key R&D Program of China (2022YFD1400700), the National Natural Science Foundation of China (U1802233, 31900420, and 32260021), the independent research fund of Yunnan Characteristic Plant Extraction Laboratory (2022YKZY006), and the Major Science and Technology Project in Yunnan Province of China (202001BC070004).

## Additional information

### Funding

| Funder | Grant reference number | Author |
|---|---|---|
| Ministry of Science and Technology of the People's Republic of China | 2022YFD1400700 | Yi-Cheng Ma |
| Foundation for Innovative Research Groups of the National Natural Science Foundation of China | U1802233 | Cheng-Gang Zou |
| Yunnan Characteristic Plant Extraction Laboratory | 2022YKZY006 | Cheng-Gang Zou |
| Foundation for Innovative Research Groups of the National Natural Science Foundation of China | 31900420 | Yi-Cheng Ma |
| Foundation for Innovative Research Groups of the National Natural Science Foundation of China | 32260021 | Yi-Cheng Ma |
| Yunnan Province of China | 202001BC070004 | Cheng-Gang Zou |

The funders had no role in study design, data collection, and interpretation, or the decision to submit the work for publication.

### Author contributions

Rui-Qiu Yang, Data curation, Software, Formal analysis, Validation, Investigation, Visualization, Methodology; Yong-Hong Chen, Data curation, Software, Formal analysis, Investigation, Visualization, Methodology; Qin-yi Wu, Data curation, Software, Formal analysis, Investigation; Jie Tang, Software, Formal analysis, Investigation, Methodology; Shan-Zhuang Niu, Software, Investigation; Qiu Zhao, Software, Formal analysis; Yi-Cheng Ma, Conceptualization, Resources, Funding acquisition, Visualization, Writing - original draft, Writing - review and editing; Cheng-Gang Zou, Conceptualization, Funding acquisition, Writing - original draft, Writing - review and editing

## Author ORCIDs

Cheng-Gang Zou http://orcid.org/0000-0001-5519-4402

## Decision letter and Author response

Decision letter https://doi.org/10.7554/eLife.85362.sa1
Author response https://doi.org/10.7554/eLife.85362.sa2

## Additional files

### Supplementary files

• MDAR checklist

• Supplementary file 1. Table S1. The $^1$H and $^{13}$C NMR spectroscopic data of indole at 600 MHz for $^1$H NMR and 150 MHz for $^{13}$C NMR with reference to the solvent signals.

• Supplementary file 2. Table S2. The data of activity-guided isolation.

### Data availability

All data generated or analyzed during this study are included in the manuscript and supporting source data file.

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
