## [Editor Report]

This fundamental study provides compelling evidence for a new mechanism of host-microbe interaction, with indole, produced by proliferating bacteria in the *C. elegans* digestive system, signalling through the host via the transcription factor DAF-16 to induce the expression of genes controlling bacterial growth in the gut. The work is relevant to a wide audience as it invites deeper research into this mechanism, while also serving as a template for similar microbiome/host interactions in other systems.

---

## [Decision Letter]

**Decision letter after peer review:**

Thank you for submitting your article “Indole produced during dysbiosis mediates host-microorganism chemical” for consideration by *eLife*. Your article has been reviewed by 3 peer reviewers, including Martin Sebastian Denzel as Reviewing Editor and Reviewer #1, and the evaluation has been overseen by Wendy Garrett as the Senior Editor.

Essential revisions:

1) How was indole identified as a biologically relevant molecule?

2) What is the role of daf-2 in the context of their work?

3) How is indole sensed? Does it signal to or interact with TRPA1?

4) Is indole taken up by the intact gut or only after leakage occurs? How might it affect DAF-16?

*Reviewer #1 (Recommendations for the authors):*

In all, this is a very convincing paper that beautifully delineates the microbiome/host cross-talk in *C. elegans*. I have a few thoughts that, if addressed, would further strengthen the paper.

How was indole identified to be the relevant molecule? How was "activity-guided isolation" done? I see in the methods that fraction 26 was identified – it would be nice to provide the data in the paper that lead to this important finding in the paper.

Dysbiosis for the purpose of this paper is defined as the aberrant growth of normal bacteria. I wonder about the daf-16 response to pathogenic bacteria. Do pathogens produce more (or less) indole? Is indole a "marker" for the presence of live bacteria in the gut, or would it be a specific response to certain bacterial strains?

The text refers to "endogenous indole": this meant indole produced by bacteria in the gut? If so, it would help to word this more specifically.

Survival statistics: I was unable to access the source data – would it be possible to specify the survival experiments in the legends or in the methods? How many survival assays were done per experiment and how many animals were used?

*Reviewer #2 (Recommendations for the authors):*

Below I detail the main issues:

1. It looks like the authors are aware of the complications of using heat-killed bacteria as a control (given the added controls in supplementary figures). Indeed, adult worms don't do well on heat-killed *E. coli* and were suggested to be starved, either energetically, or for certain essential nutrients. All experiments would have been easier to interpret if paraformaldehyde-killed bacteria were used as controls (see Beydoun et al., 2021). Nevertheless, the authors did include controls with antibiotic-killed bacteria, and with starvation per se, to demonstrate that starvation is not involved, and that baseline established for comparisons is similar in worms raised on bacteria killed not by heat (method describing how ampicillin was employed to kill bacteria should be included, as well as a note on whether bacteria were dead or only metabolically inactive). The additional control experiments, currently presented in Figure 1- supp. 1 and 2, should be incorporated into the main Figure 1.

2. Throughout the paper, the authors used SEMs to describe the variance in their data. SDs are more appropriate for this.

3. The authors seem to suggest that the sensing of indole is inside the gut, released by accumulating bacteria. This leaves the question of how such sensing occurs unanswered. As far as I know, there is no knowledge about neuronal sensing of gut conditions. This should be addressed. The authors demonstrate tight correlation between gut bacteria CFUs and indole levels (Figure 2- supp. 3), but given the ability of supplemented external indole (more info is required about how this is carried out) to increase internal indole concentrations, I don't see why should such correlation exist – there are two sources for gut indole levels – colonizing bacteria and external bacteria, so excess environmental availability should drown any correlation.

Perhaps modulating bacterial accumulation (as in grinder defective tnt-3 mutants), could help clarify that.

4. The importance of TRPA-1 for indole-induced activation of DAF-16 is central to the paper. To provide stronger support, the authors may want to replicate the phenotypes, shown in Figure 4 with trpa-1 RNAi, also in trpa-1 null mutants.

Also, are the effects of TRPA-1 and DAF-16 KO/KD additive or not? This is important to establish the axis suggested here.

5. Lack of SKN-1 nuclear localization does not strike me as the most conclusive evidence for lack of p38 activation by indole. For one, p38 also activates ATF-7. The authors should also run qRT-PCR for p38 targets, for example T24B8.5, or use available GFP reporter strains for such targets.

6. The authors should also examine the outcome of activating TRPA-1 with its previously described agonist podacarpic acid (which they mention). Does it activate DAF-16 nuclear localization? Seems to be simple enough.

For discussion: do the authors consider indole as TRPA-1 agonist? Is TRPA-1 a ligand-activated channel? Have homologs been shown in other organisms to have more than one agonist, with different outcomes? How similar are the structures of podacarpic acid and indole?

7. Given the many targets of DAF-16, and the redundant nature of innate immune protection, how do the authors explain the ability of lys-7 and lys-8 to account for the full contribution of DAF-16 to lifespan and OP50 control?

---

## [Author Response]

Essential revisions:1) How was indole identified as a biologically relevant molecule?

We appreciate the concerns of the reviewing editor. Activity-guided isolation was

performed as follows: The crude extract of *E. coli* supernatant metabolites was divided into 45 fractions according to polarity using Ultimate 3000 HPLC (Thermofisher, Waltham, MA) coupled with automated fraction collector. After freeze-drying each fraction, 1 mg of metabolites were dissolved in DMSO for DAF-16 nuclear localization assay in worms (Please see new Supplementary Table S2). The 26th fraction with DAF-16 nuclear translocation-inducing activity was then separated on silica gel column (200-300 mesh) with a continuous gradient of decreasing polarity (100%, 70%, 50%, 30%, petroleum ether/acetone) to yield four fractions (26a-d). Only the fraction of 26b could induce DAF-16 nuclear translocation. Then the fraction was further separated using a Sephadex LH-20 column to yield 32 fractions. The 26b-11th fraction with DAF-16 nuclear translocation-inducing activity contained a single compound identified by thin layer chromatography, mass spectrometry and nuclear magnetic resonance (NMR). The compound exhibited a quasimolecular ion peak at m/z 181.0782 [M+H]+ in the positive APCI-MS, and was assigned to a molecular formula of C8H7N. A comparison of these 1H NMR and 13C NMR spectra with the data reported in the literature revealed that the compound was indole (Yagudaev, 1986). The detailed information was provided in our revised version.

2) What is the role of daf-2 in the context of their work?

We appreciate the concerns of the reviewing editor. It has been shown that DAF-2

initiates a kinase cascade that leads to the phosphorylation and cytoplasmic retention of DAF-16. By contrast, a reduction in the DAF-2 signaling leads to the dephosphorylation of DAF-16, allowing its nuclear translocation. We found that the mRNA levels of daf-2 were significantly increased in worms on days 4 and 7 in the presence of either live or dead *E. coli* OP50, compared with those in worms on day 1 (Author response image 1). The nuclear translocation of DAF-16 in the intestine was increased in worms fed live *E. coli* OP50 on days 4 and 7, but not in age-matched WT worms fed heat-killed (HK) *E. coli* OP50 or ampicillin-killed *E. coli* OP50 (Figure 1A-1C). In addition, supplementation with indole significantly induced the nuclear translocation of DAF-16 (Figure 2B), but did not alter the mRNA levels of daf-2 in young adult worms (Author response image 1). To conclude, the activation of DAF-16 is independent of DAF-2 during C. elegans aging.

**Author response image 1. sa2fig1:** DAF-16 nuclear translocation is independent of DAF-2. (A) The mRNA levels of *daf-2* were gradually increased in worms with age. ***P* < 0.01; ****P* < 0.001; ns, not significant. (B) The mRNA levels of *daf-2* were not altered after treatment with indole for 24 hours. ns, not significant.

3) How is indole sensed? Does it signal to or interact with TRPA1?

We thank the reviewing editor for raising this important issue. The transporter of

indole has not been identified in *C. elegans*. In Arabidopsis, ATP-binding cassette (ABC) transporter G family 37(ABCG37) has been reported to transport a range of indole derivatives (Ruzicka et al., 2010). However, all fifteen C. elegans ABC transporters share less than 30% sequence identity with ABCG37, making it difficult to determine which one is the transporter for indole and indole derivatives (Author response image 2). Recently, Ye et al. have demonstrated that indole and indole-3-carboxaldehyde (IAld) are agonists of TRPA1, which is conserved in vertebrates (Ye et al., 2021). Thus, it is mostly likely that indole acts as an agonist of TRPA-1 in C. elegans by directly binding to TRPA-1.

**Author response image 2. sa2fig2:** Sequence alignment between *Arabidopsis* ABCG37 and 15 *C. elegans* ABC transporters.

4) Is indole taken up by the intact gut or only after leakage occurs? How might it affect DAF-16?

We appreciate the concerns of the reviewing editor. Although the transporter of indole in *C. elegans* remains to be characterized, we exclude a possible mechanism by which an increase in the indole levels in worms is associated with gut leakage. We found that supplementation with indole significantly increased the indole levels in young adult worms on day 1 (Figure 2—figure supplement 3B), and induced the nuclear translocation of DAF-16 in worms (Figure 2B). The intestine barrier is intact in 1-day-old worms (Figure 1I). Thus, the indole is taken up by the intact gut.

We speculate that the activation of TRPA-1 by indole in neurons could induce a pathway that may release a neurotransmitter, thereby leading to the activation of DAF-16 in the intestine.

Reviewer #1 (Recommendations for the authors):In all, this is a very convincing paper that beautifully delineates the microbiome/host cross-talk in *C. elegans*. I have a few thoughts that, if addressed, would further strengthen the paper.How was indole identified to be the relevant molecule? How was "activity-guided isolation" done? I see in the methods that fraction 26 was identified – it would be nice to provide the data in the paper that lead to this important finding in the paper.

We appreciate the concerns of the reviewer. Activity-guided isolation was performed as follows: The crude extract of *E. coli* supernatant metabolites was divided into 45 fractions according to polarity using Ultimate 3000 HPLC (Thermofisher, Waltham, MA) coupled with automated fraction collector. After freeze-drying each fraction, 1 mg of metabolites were dissolved in DMSO for DAF-16 nuclear localization assay in worms (Please see new Supplementary Table S2). The 26th fraction with DAF-16 nuclear translocation-inducing activity was then separated on silica gel column (200-300 mesh) with a continuous gradient of decreasing polarity (100%, 70%, 50%, 30%, petroleum ether/acetone) to yield four fractions (26a-d). Only the fraction of 26b could induce DAF-16 nuclear translocation. Then the fraction was further separated using a Sephadex LH-20 column to yield 32 fractions. The 26b-11th fraction with DAF-16 nuclear translocation-inducing activity contained a single compound identified by thin layer chromatography, mass spectrometry and nuclear magnetic resonance (NMR). The compound exhibited a quasimolecular ion peak at m/z 181.0782 [M+H]+ in the positive APCI-MS, and was assigned to a molecular formula of C8H7N. A comparison of these 1H NMR and 13C NMR spectra with the data reported in the literature revealed that the compound was indole (Yagudaev, 1986). The detailed information was provided in our revised version.

Dysbiosis for the purpose of this paper is defined as the aberrant growth of normal bacteria. I wonder about the daf-16 response to pathogenic bacteria. Do pathogens produce more (or less) indole? Is indole a "marker" for the presence of live bacteria in the gut, or would it be a specific response to certain bacterial strains?

In response to the reviewer’s suggestion, we tested three pathogenic bacteria (*Salmonella enterica serovar Typhimurium*, *Pseudomonas aeruginosa*, and *Staphylococcus aureus*). We found that the nuclear translocation of DAF-16 was also increased in the intestine of worms fed *S. Typhimurium* 468, but not *P. aeruginosa* PA14 and *S. aureus* ATCC 25923 (Author response image 3). Furthermore, we measured the levels of indole in the worms fed with each of these pathogenic bacteria. We found that the levels of indole were 80.4, 25.5, 22.8, and 23.1 nmol/g dry weight in worms fed live *E. coli* OP50, *S. Typhimurium* 468, *P. aeruginosa* PA14, and *S. aureus* ATCC 25923 on day 4, respectively (Author response image 3). In addition, the levels of indole were 426.2, 0.7, 1.1, and 21.9 μmol/L in the culture

supernatants of *E. coli*, *S. typhimurium*, *P. aeruginosa*, and *S. aureus*, respectively (Author response image 3). Based on these observations, we conclude that indole-induced DAF-16 activation is a specific response to *E. coli*.

**Author response image 3. sa2fig3:** Production of indole is associated with induction of DAF-16 nuclear translocation in *E. coli*, not in other three bacteria. (A) The nuclear translocation of DAF-16::GFP was increased in the intestine of worms fed *E. coli* and *S. Typhimurium,* but not *P. aeruginosa* PA14 or *S. aureus* on day 4. These results are means ± SD of three independent experiments (n > 35 worms per experiment). ****P* < 0.001. (B) The levels of indole in worms fed *S. Typhimurium, P. aeruginosa,* and *S. aureus* were much lower than those in worms fed *E. coli* OP50 on day 4. These results are means ± SD of three independent experiments. **P* < 0.05. (C) The levels of indole in the culture supernatants of *S. Typhimurium, P. aeruginosa*, and *S. aureus* were much lower than those of *E. coli* OP50. ****P* < 0.001.

The text refers to "endogenous indole": this meant indole produced by bacteria in the gut? If so, it would help to word this more specifically.

We appreciate the concerns of the reviewer. Yes, endogenous indole we described here means “indole produced by bacteria in the gut”. We have changed the “endogenous indole” to “indole produced by bacteria in the gut” in the revised manuscript.

Survival statistics: I was unable to access the source data – would it be possible to specify the survival experiments in the legends or in the methods? How many survival assays were done per experiment and how many animals were used?

Lifespan data are collected in source data.

Reviewer #2 (Recommendations for the authors):Below I detail the main issues:1. It looks like the authors are aware of the complications of using heat-killed bacteria as a control (given the added controls in supplementary figures). Indeed, adult worms don't do well on heat-killed *E. coli* and were suggested to be starved, either energetically, or for certain essential nutrients. All experiments would have been easier to interpret if paraformaldehyde-killed bacteria were used as controls (see Beydoun et al., 2021). Nevertheless, the authors did include controls with antibiotic-killed bacteria, and with starvation per se, to demonstrate that starvation is not involved, and that baseline established for comparisons is similar in worms raised on bacteria killed not by heat (method describing how ampicillin was employed to kill bacteria should be included, as well as a note on whether bacteria were dead or only metabolically inactive). The additional control experiments, currently presented in Figure 1- supp. 1 and 2, should be incorporated into the main Figure 1.

In response to the reviewer’s suggestion, the details of ampicillin was employed to

kill bacteria were provided in the section of Methods. Meanwhile, we incorporated the previous Figure 1—figure supplement 1 and 2 into Figure 1.

2. Throughout the paper, the authors used SEMs to describe the variance in their data. SDs are more appropriate for this.

Amended.

3. The authors seem to suggest that the sensing of indole is inside the gut, released by accumulating bacteria. This leaves the question of how such sensing occurs unanswered. As far as I know, there is no knowledge about neuronal sensing of gut conditions. This should be addressed. The authors demonstrate tight correlation between gut bacteria CFUs and indole levels (Figure 2- supp. 3), but given the ability of supplemented external indole (more info is required about how this is carried out) to increase internal indole concentrations, I don't see why should such correlation exist – there are two sources for gut indole levels – colonizing bacteria and external bacteria, so excess environmental availability should drown any correlation.Perhaps modulating bacterial accumulation (as in grinder defective tnt-3 mutants), could help clarify that.

We appreciate the concerns of the reviewer. Reviewer #3 also pointed out this problem. In this study, our data showed that the levels of indole were 30.9, 71.9, and 105.9 nmol/g dry weight in worms fed live E. coli OP50 on days 1, 4, and 7, respectively (Figure 2C). This increase in the levels of indole in worms was accompanied by an increase in CFU of live E. coli OP50 in the intestine of worms with age (Figure 2C). In addition, we determined the levels of indole in worms fed HK E. coli OP50, and found that the levels of indole were 28.2, 31.6, and 36.1 nmol/g dry weight in worms fed HK E. coli OP50 on days 1, 4, and 7, respectively (Figure 2—figure supplement 3A). It should be noted that the levels of indole in worms fed dead E. coli OP50 on day 1 were comparable of those in worms fed live E. coli OP50 on day 1 (30.9 vs 28.2 nmol/g dry weight). However, the levels of indole were not increased in worms fed HK E. coli OP50 on days 4 and 7. Furthermore, the observation that DAF-16 was retained in the cytoplasm of the intestine in worms fed live E. coli OP50 on day 1 (Figure 1A and 1B) also indicates that indole produced by *E. coli* OP50 on the NGM plates is not enough to induce DAF-16 nuclear translocation. By contrast, supplementation with indole (50-200 μM) significantly increased the indole levels in worms on day 1 (Figure 2—figure supplement 3B), which could induce the nuclear translocation of DAF-16 in worms (Figure 2B). Thus, the increase in the levels of indole in worms with age results from intestinal accumulation of live E. coli OP50, rather than indole produced by E. coli OP50 on the NGM plates.

4. The importance of TRPA-1 for indole-induced activation of DAF-16 is central to the paper. To provide stronger support, the authors may want to replicate the phenotypes, shown in Figure 4 with trpa-1 RNAi, also in trpa-1 null mutants.Also, are the effects of TRPA-1 and DAF-16 KO/KD additive or not? This is important to establish the axis suggested here.

We appreciate the concerns of the reviewer and have performed the suggested experiments. We found that supplementation with indole no longer extended the lifespan of *trpa-1(ok999)* mutants (new Figure 4—figure supplement 2A).

Furthermore, knockdown of *daf-16* by RNAi did not further shorten the lifespan of *trpa-1(ok999)* mutants (new Figure 4—figure supplement 2A). Moreover, supplementation with indole failed to suppress the accumulation of *E. coli* K-12 strain in *trpa-1(ok999)* mutants subjected to either empty vector or *daf-16* RNAi (new Figure 4—figure supplement 2B). These results suggest that indole exhibits its function in extending the lifespan of worms and inhibiting the bacterial accumulation primarily via TRPA-1-DAF-16 axis.

5. Lack of SKN-1 nuclear localization does not strike me as the most conclusive evidence for lack of p38 activation by indole. For one, p38 also activates ATF-7. The authors should also run qRT-PCR for p38 targets, for example T24B8.5, or use available GFP reporter strains for such targets.

We appreciate the concerns of the reviewer and have performed the suggested experiments. We tested the effect of indole on five p38 target genes (*T24B8.5*, *F35e12.5*, *nlp-29, Y37a1a.2,* and *F08g5.6*). We found that supplementation with indole only increased the mRNA level of *Y37a1a.2*, but not the other four genes (Author response image 4). These results suggest that indole treatment does not activate the PMK-1/p38 MAPK pathway.

**Author response image 4. sa2fig4:** The effect of indole on the expression of p38 target genes. Supplementation with indole only upregulated the expression of *Y37a1a.2*, but not *T24B8.5*, *F35e12.5*, *nlp-29,* and *F08g5.6.* **P* < 0.05.

6. The authors should also examine the outcome of activating TRPA-1 with its previously described agonist podacarpic acid (which they mention). Does it activate DAF-16 nuclear localization? Seems to be simple enough.For discussion: do the authors consider indole as TRPA-1 agonist? Is TRPA-1 a ligand-activated channel? Have homologs been shown in other organisms to have more than one agonist, with different outcomes? How similar are the structures of podacarpic acid and indole?

We appreciate the concerns of the reviewer and have performed the suggested

experiments. It has been shown that podocarpic acid can activate SKN-1 via TRPA-1 (Chaudhuri et al., 2016). We found that supplementation with 0.02 mM podocarpic acid failed to induce the nuclear translocation of DAF-16 in worms on day 1 (new Figure 4-figure supplement 3C). Xiao et al. have demonstrated that TRPA-1 activated by low temperatures mediates calcium influx, which in turn stimulates the PKC-2-SGK-1 signaling to promote the transcription activity of DAF-16 (Xiao et al., 2013). This suggests that TRPA-1 activation by podocarpic acid, low temperature and indole occurs largely through distinct mechanisms.

Recently, Ye et al. have shown that both indole and indole-3-carboxaldehyde (IAld) are TRPA1 agonists in vertebrates (Ye et al., 2021). The agonists of mammalian TRPA1 include both non-electrophilic and electrophilic agonists, such as icilin, carvacrol, docosahexaenoic acid (DHA), isoflurane, allyl isothiocyanate (AITC), acrolein, umbellulone, and etodolac (Nishida, Kuwahara, Kozai, Sakaguchi, and Mori, 2015). Activated TRPA1 have different effects in mammals. For example, TRPA1 induces the releasing of calcitonin gene-related peptide in perivascular sensory nerves, which in turn binds corresponding G protein-coupled receptor, and causes membrane hyperpolarization and arterial dilation on smooth muscle cells (Talavera et al., 2020). Additionally, the activation of TRPA1 by indole and IAld induces the secretion of the neurotransmitter serotonin in zebrafish (Ye et al., 2021).

Both Indole and podocarpic acid are non-electrophilic agonists of TAPR-1, the structures of podocarpic acid and indole are totally different. Podocarpic acid is an abietane diterpenoid lacking the isopropyl substituent with an aromatic C-ring and a hydroxy group at the 12-position (W.Campbell, 1942). Indole is an electron-rich aromatic system containing enamine embedded C2-C3 π bond and strong nucleophilic C3 carbon (Zhang et al., 2021).

In response to the reviewer’s suggestion, we added sentences “Recently, Ye et al. have shown that both indole and indole-3-carboxaldehyde (IAld) are TRPA1 agonists in vertebrates (Ye et al., 2021). As the agonist of TRPA-1, podocarpic acid can activate SKN-1 via TRPA-1 in worms (Chaudhuri et al., 2016). However, podocarpic acid fails to trigger DAF-16 translocation. Unlike podocarpic acid, indole cannot activate SKN-1 in worms. Although both Indole and podocarpic acid are non-electrophilic agonists of TAPR-1, the structures of podocarpic acid and indole are totally different. These results suggest that TRPA-1 activation by podocarpic acid, low temperature, and indole occurs largely through distinct mechanisms.” in the section of Discussion.

7. Given the many targets of DAF-16, and the redundant nature of innate immune protection, how do the authors explain the ability of lys-7 and lys-8 to account for the full contribution of DAF-16 to lifespan and OP50 control?

We appreciate the concerns of the reviewer. *C. elegans* possesses a variety of putative antimicrobial proteins, such as lysozymes, defensin-like peptides, neuropeptide-like proteins, and caenacins (Dierking, Yang, and Schulenburg, 2016), Of these antimicrobial proteins, the expressions of lysozyme genes, such as *lys-2*, *lys-7*, and *lys-8*, are markedly up-regulated in 4-day-old worms, which is dependent on DAF-16 (Li et al., 2019). Although we have demonstrated that LYS-7 and LYS-8 are required for normal lifespan, we do not exclude the role for other targets of DAF-16 in maintaining normal lifespan in worms.

References

Cabreiro, F., and Gems, D. (2013). Worms need microbes too: microbiota, health and aging in *Caenorhabditis elegans*. EMBO Mol Med, 5(9), 1300-1310.

Chaudhuri, J., Bose, N., Gong, J., Hall, D., Rifkind, A., Bhaumik, D., Kapahi, P. (2016). A *Caenorhabditis elegans* Model Elucidates a Conserved Role for TRPA1-Nrf Signaling

in Reactive α-Dicarbonyl Detoxification. Curr Biol, 26(22), 3014-3025.

Dierking, K., Yang, W., and Schulenburg, H. (2016). Antimicrobial effectors in the nematode *Caenorhabditis elegans*: an outgroup to the Arthropoda. Philos Trans R Soc Lond B Biol Sci, 371(1695).

Lee, R. Y., Hench, J., and Ruvkun, G. (2001). Regulation of *C. elegans* DAF-16 and its human ortholog FKHRL1 by the daf-2 insulin-like signaling pathway. Curr Biol, 11(24), 1950-1957.

Li, S. T., Zhao, H. Q., Zhang, P., Liang, C. Y., Zhang, Y. P., Hsu, A. L., and Dong, M. Q. (2019).

DAF-16 stabilizes the aging transcriptome and is activated in mid-aged *Caenorhabditis elegans* to cope with internal stress. Aging Cell, 18(3), e12896.

Lin, K., Hsin, H., Libina, N., and Kenyon, C. (2001). Regulation of the *Caenorhabditis elegans* longevity protein DAF-16 by insulin/IGF-1 and germline signaling. Nat Genet, 28(2), 139-145.

Nishida, M., Kuwahara, K., Kozai, D., Sakaguchi, R., and Mori, Y. (2015). TRP Channels: Their Function and Potentiality as Drug Targets. In K. Nakao, N. Minato and S.

Uemoto (Eds.), Innovative Medicine: Basic Research and Development (pp. 195-218).

Tokyo.

Ruzicka, K., Strader, L. C., Bailly, A., Yang, H., Blakeslee, J., Langowski, L.,... Friml, J. (2010). Arabidopsis PIS1 encodes the ABCG37 transporter of auxinic compounds

including the auxin precursor indole-3-butyric acid. Proc Natl Acad Sci U S A, 107(23), 10749-10753.

Sonowal, R., Swimm, A., Sahoo, A., Luo, L., Matsunaga, Y., Wu, Z.,... Kalman, D. (2017). Indoles from commensal bacteria extend healthspan. Proc Natl Acad Sci U S A, 114(36), E7506-E7515.

Talavera, K., Startek, J. B., Alvarez-Collazo, J., Boonen, B., Alpizar, Y. A., Sanchez, A.,...

Nilius, B. (2020). Mammalian Transient Receptor Potential TRPA1 Channels: From Structure to Disease. Physiol Rev, 100(2), 725-803.

W.Campbell, D. T. (1942). The Structure and Configuration of Resin Acids. Podocarpic Acid and Ferruginol. J Am Chem Soc, 64(4), 928-935.

Winston, W. M., Molodowitch, C., and Hunter, C. P. (2002). Systemic RNAi in *C. elegans*

requires the putative transmembrane protein SID-1. Science, 295(5564), 2456-2459.

Xiao, R., Zhang, B., Dong, Y., Gong, J., Xu, T., Liu, J., and Xu, X. Z. (2013). A genetic

program promotes *C. elegans* longevity at cold temperatures via a thermosensitive TRP channel. Cell, 152(4), 806-817.

Yagudaev. (1986). Application of H and C NMR spectroscopy in structural investigations of Vinca indole alkaloids. Chemistry of Natural Compounds, 22(1), 1-13.

Ye, L., Bae, M., Cassilly, C. D., Jabba, S. V., Thorpe, D. W., Martin, A. M.,... Rawls, J. F. (2021). Enteroendocrine cells sense bacterial tryptophan catabolites to activate enteric and vagal neuronal pathways. Cell Host Microbe, 29(2), 179-196 e179.

Zhang, Y., Ji, P., Gao, F., Dong, Y., Huang, H., Wang, C., Wang, W. (2021). Organophotocatalytic dearomatization of indoles, pyrroles and benzo(thio)furans via

a Giese-type transformation. Commun Chem, 4(1), 20.

Zygmunt, P. M., and Hogestatt, E. D. (2014). Trpa1. Handb Exp Pharmacol, 222, 583-630.